# RD-HRL: Generating Reliable Sub-Goals for Long-Horizon Sparse-Reward Tasks

**Yixiang Shan**
School of Artificial Intelligence, International Center of Future Science
Jilin University
China
shanyx20@mails.jlu.edu.cn

**Haipeng Liu, Ting Long** *
School of Artificial Intelligence
Jilin University
China
liuhp22@mails.jlu.edu.cn, longting@jlu.edu.cn

**Yi Chang** *
School of Artificial Intelligence, International Center of Future Science
Jilin University
Engineering Research Center of Knowledge-Driven Human-Machine Intelligence
Ministry of Education
China
yichang@jlu.edu.cn

## Abstract

Long-horizon sparse-reward tasks, such as goal-conditioned or robot manipulation tasks, remain challenging in offline reinforcement learning due to the credit assignment problem. Hierarchical methods have been proposed to tackle this problem by introducing sub-goal planning guided by value functions, which in principle can shorten the effective planning horizon for both high-level and low-level planners, and thereby avoiding the credit assignment problem. However, we demonstrate that the sub-goal selection mechanism is unreliable, as it relies on value functions suffering from generalization noise, which misguides value estimation and thus leads to sub-optimal sub-goals. In this work, to provide more reliable sub-goals, we novelly introduce a reliability-driven decision mechanism, and propose Reliability-Driven HRL (RD-HRL) as the solution. The reliability-driven decision mechanism provide decision-level targets for high-level policy, thereby providing noise-immune decision spaces for them, ensuring the reliability of sub-goals (which are termed as action-level targets in this paper). Comprehensive experimental results demonstrate that our approach RD-HRL outperforms baseline methods across multiple benchmarks, highlighting the competitive advantages of RD-HRL. Our code is anonymously available at https://github.com/Looomo/RD-HRL-public.

## 1 Introduction

Reinforcement Learning (RL) (Kaelbling et al., 1996; Wiering and Van Otterlo, 2012; Li, 2017; Sutton et al., 1999) has achieved remarkable success in many tasks, but it still faces significant challenges in long-horizon sparse-rewards situations (Lee et al., 2022; Shin and Kim, 2023; Wang et al., 2020). In such tasks, the agent often receives meaningful feedback only upon reaching distant goals, which makes credit assignment and exploration particularly difficult (Zhou et al., 2020; Pignatelli et al., 2023). A common solution is Hierarchical Reinforcement Learning (HRL) (Pateria

---

*Corresoponding Authors.

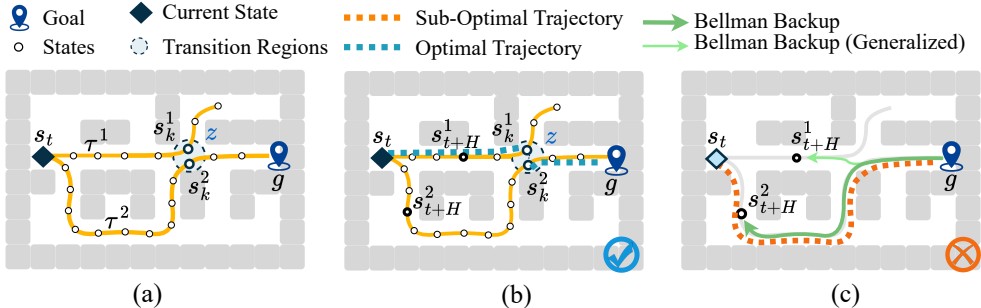

Figure 1: The dataset contains two trajectories, $\boldsymbol{\tau}_1$ and $\boldsymbol{\tau}_2$, as is shown in (a). Note that, $\boldsymbol{\tau}_2$ successfully reached the goal point $\boldsymbol{g}$, yet $\boldsymbol{\tau}_2$ contains a sub-optimal sub-trajectory; meanwhile, $\boldsymbol{\tau}_1$ fails to reach the goal, but $\boldsymbol{\tau}_1$ provides an optimal sub-trajectory from $\boldsymbol{s}_t$ to $\boldsymbol{s}_k^1$. In (b), we show the optimal trajectory with blue dotted line, which contains the key transition $\{\ \boldsymbol{s}_k^1 \to \boldsymbol{s}_k^2\ \}$; in (c), we show that existing HRL methods tend to plan sub-optimal trajectories (the orange dotted line) and the attenuate generalized Bellman backup.

et al., 2021), which decomposes complex tasks into a hierarchy of sub-tasks to address the issue. Specifically, HRL is composed of a high-level policy and a low-level policy, where the high-level policy proposes intermediate sub-goals with a value function, while the low-level policy learns how to reach these sub-goals. By structuring the decision-making process in stages, HRL alleviates the learning difficulties faced by agents in long-horizon sparse-reward situations (Barto and Mahadevan, 2003; Nachum et al., 2018).

However, the value function which is used to propose sub-goals is often subject to generalization errors in practice, leading to unreliable sub-goals. Consider a simple scenario presented in Figure 1 (a), where the agent starts from state $\boldsymbol{s}_t$ and aims to reach the goal $\boldsymbol{g}$. Intuitively, the optimal path should include $\boldsymbol{s}_{t+H}^1$, as is shown in the blue dotted line from Figure 1 (b). However, the value estimation of $\boldsymbol{s}_{t+H}^1$ relies on cross-trajectory Bellman backup (*i.e.*, generalized Bellman backup from region $z = \{\boldsymbol{s}_k^1, \boldsymbol{s}_k^2\}$), where the generalized signal is often attenuated and unreliable (Zhang et al., 2024), as is shown in Figure 1 (c). This may leads to the underestimated value of the optimal sub-goal $\boldsymbol{s}_{t+H}^1$, leading to the high-level policy to select a suboptimal sub-goal $\boldsymbol{s}_{t+H}^2$ instead, which eventually results in a suboptimal trajectory, as is shown in the orange dotted line in Figure 1 (c). For the sake of simplicity, we designate regions analogous to $z$, which facilitate the connection across different trajectories, as transition regions.

Intuitively, if we can prevent the high-level policy from comparing sub-goal candidates whose value signals are unreliable, and restrict its decision space to local regions that do not require such generalization, the impact of generalization error can be substantially reduced. Building on this insight, we propose Reliability-Driven HRL (RD-HRL), which augments HRL with a reliability-driven decision mechanism. The reliability-driven decision mechanism selects decision-level targets for the high-level policy from transition regions $\mathcal{Z} = \{z\}$, thereby confining the high-level decision space to areas without generalization requirements and decoupling sub-goal selection from cross-trajectory value estimates. Note that for better presentation, we define the sub-goals generated by the reliability-driven mechanism for the high-level policy as decision-level target, and define the sub-goals assigned by the high-level policy to the low-level policy as action-level target. In this way, we decompose action-level target (*i.e.*, sub-goals) planning into two reliable subproblems: (1) providing suitable decision-level targets through the reliability-driven decision mechanism, and (2) producing reliable action-level targets conditioned on decision-level targets.

Specifically, the reliability-driven decision mechanism is composed of the Transition Region Extraction (TRE) module, the Target Identification (TI) module and the Target Evaluation (TE) module. The TRE module filters transition regions from the offline dataset, providing the candidates of decision-level target for the TI module; the TE module estimates the low-noise value of transition regions for the TI module; the TI module selects proper transition regions as decision-level targets for the high-level policy with the help of the TE module. Combining the reliability-driven

decision mechanism with the HRL framework, we propose RD-HRL for better decision-making in long-horizon sparse-reward tasks. In summary, the contribution of our work can be summarized as:

- We propose a method **Reliability-Driven HRL (RD-HRL)**, which novelly augments HRL with a reliability-driven decision mechanism to provide decision-level target from transition regions for the high-level policy, thereby disentangling the high-level policy from generalization error, providing reliable action-level target (*i.e.*, sub-goals).

- We propose the Target Evaluation (TE) module for reliable decision-level target generation, and theoretically proved that our proposed TE module can effectively reduce value noise in long-horizon sparse-reward scenarios. Experimental results further demonstrate the importance of the Target Evaluation (TE) module.

- We conduct extensive experiments to validate the effectiveness of RD-HRL, along with in-depth ablation studies to verify the impact of its key designs.

## 2 PRELIMINARIES

### 2.1 OFFLINE GOAL-CONDITIONED REINFORCEMENT LEARNING

Offline goal-conditioned reinforcement learning (offline GCRL) is the most representative task in long-horizon sparse-reward scenarios, which can be formulated as a Markov Decision Process (MDP) (Sutton and Barto, 2018) $\mathcal{M} = \{\mathcal{S}, \mathcal{A}, \mathcal{P}, \gamma, \mathcal{R}, \boldsymbol{g}\}$ with a dataset $\mathcal{D}$, where $\mathcal{S}$ is the state space, $\mathcal{A}$ is the action space, $\mathcal{P}(\boldsymbol{s}_{t+1}|\boldsymbol{s}_t, \boldsymbol{a}_t)$ is the transition function, $\gamma$ represents the discount factor, $\boldsymbol{g}$ represents the goal, $\mathcal{R}(\boldsymbol{s}_t, \boldsymbol{a}_t, \boldsymbol{g})$ is a goal-conditioned reward function, which is commonly formulated as:

$$r_t = \mathcal{R}(\boldsymbol{s}_t, \boldsymbol{a}_t, \boldsymbol{g}) = \begin{cases} 1, & ||\phi(\boldsymbol{s}_t) - \boldsymbol{g}||_2 \leq \epsilon, \\ 0, & \text{otherwise}, \end{cases} \tag{1}$$

where $r_t$ represents the reward received at time $t$, $\epsilon$ represents a given threshold, $\phi(\cdot)$ maps a state from the state space to the goal space. Following Liu et al. (2022), we assume the goal space is identical to the state space, thus we omit $\phi(\cdot)$ in this paper.

At each step $t$, the agent responds to the state of the environment $\boldsymbol{s}_t$ and goal $\boldsymbol{g}$ by action $\boldsymbol{a}_t$ according to policy $\pi_\theta$ parameterized by $\theta$, and gets an instant reward $r_t$. The interaction history is formulated as a trajectory $\boldsymbol{\tau} = \{(\boldsymbol{s}_t, \boldsymbol{a}_t, r_t)|t \geq 0\}$, which further consists $\mathcal{D}$ as $\mathcal{D} \triangleq \{(\boldsymbol{s}_t, \boldsymbol{a}_t, r_t, \boldsymbol{s}_{t+1})|t \geq 0\}$. Our goal is learning $\pi_\theta$ to maximize the expected discounted accumulated reward without directly interacting with the environment, *i.e.*,

$$\pi_\theta = \arg\max_\theta \mathbb{E}_{\boldsymbol{\tau} \sim \pi_\theta}[\sum_{t \geq 0} \gamma^t r_t] \, . \tag{2}$$

### 2.2 HIERARCHICAL REINFORCEMENT LEARNING

Hierarchical Reinforcement Learning (HRL) has been widely used for long-horizon sparse-reward tasks, which generally consists of a high-level policy $\pi^h_{\theta_h}(\boldsymbol{g}^h_t|\boldsymbol{s}_t, \hat{\boldsymbol{g}})$ and a low-level policy $\pi^l_{\theta_l}(\boldsymbol{a}_t|\boldsymbol{s}_t, \boldsymbol{g}^h_t)$, in which $\hat{\boldsymbol{g}}$ represents the goal for the high-level policy, $\pi^h$ generates action-level targets that are easier to achieve for $\pi^l$, and $\pi^l$ provides the action to be taken. Under the general setting, the task goal $\boldsymbol{g}$ is taken as $\hat{\boldsymbol{g}}$. Formally, $\pi^h_{\theta_h}(\boldsymbol{g}^h_t|\boldsymbol{s}_t, \hat{\boldsymbol{g}})$ and $\pi^l_{\theta_l}(\boldsymbol{a}_t|\boldsymbol{s}_t, \boldsymbol{g}^h_t)$ can be learned with advantage weighting regression (AWR) objectives (Qing et al., 2024; Wang et al., 2023; Eysenbach et al., 2023; Park et al., 2024a; Osa et al., 2019; Pateria et al., 2021; Bai et al., 2024):

$$\mathcal{L}_{\theta_h} = \mathbb{E}[\exp(\beta \cdot A^h(\boldsymbol{s}_{t+H}, \boldsymbol{s}_t, \hat{\boldsymbol{g}})) \cdot \log\pi^h_{\theta_h}(\boldsymbol{s}_{t+H}|\boldsymbol{s}_t, \hat{\boldsymbol{g}})], \tag{3}$$

$$\mathcal{L}_{\theta_l} = \mathbb{E}[\exp(\beta \cdot A^l(\boldsymbol{a}_t, \boldsymbol{s}_t, \boldsymbol{s}_{t+H})) \cdot \log\pi^l_{\theta_l}(\boldsymbol{a}_t|\boldsymbol{s}_t, \boldsymbol{s}_{t+H})], , \tag{4}$$

where $A$ represents the advantage signal, and $A^h(\boldsymbol{s}_{t+H}, \boldsymbol{s}_t, \hat{\boldsymbol{g}})$ is approximated as $V(\boldsymbol{s}_{t+H}, \hat{\boldsymbol{g}}) - V(\boldsymbol{s}_t, \hat{\boldsymbol{g}})$, $A^l(\boldsymbol{a}_t, \boldsymbol{s}_t, \boldsymbol{s}_{t+H})$ is approximated as $V(\boldsymbol{s}_{t+1}, \boldsymbol{s}_{t+H}) - V(\boldsymbol{s}_t, \boldsymbol{s}_{t+H})$, $V(\boldsymbol{s}_t, \hat{\boldsymbol{g}})$ is the value of $\boldsymbol{s}_t$ conditioned on goal $\hat{\boldsymbol{g}}$, $\beta$ represents the subsumed temperature (Park et al., 2024a) , $H$ is a hyper-parameter, which is also known as waysteps. As discussed in Section 1, existing HRL methods relies on value functions affected by generalization noise to select action-level targets, which makes the optimal action-level targets challenging (Park et al., 2024a; Liu et al., 2022).

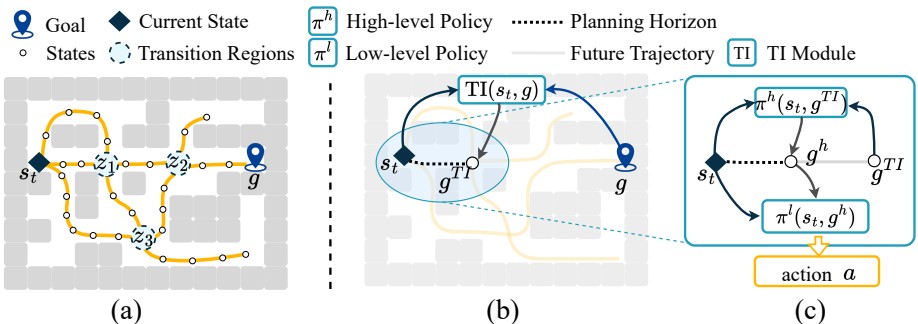

Figure 2: Framework of RD-HRL. (a) Overview of datasets, and transition regions filtered by the Transition Region Extraction (TRE) module. (b) The Transition Identification (TI) module provide $\boldsymbol{g}^{TI}$ as the decision-level target; (c) The high-level policy and low-level policy. During evaluating, the TI module predicts $z_1$ as the decision-level target $\boldsymbol{g}^{TI}$, as shown in (b); then, $\boldsymbol{g}^{TI}$ is provided to the high-level policy $\pi^h$ to generate the action-level target for the low-level policy $\pi^l$.

## 3 METHODOLOGY

To improve the action-level target selection and enhance performance in long-horizon sparse-reward tasks, we propose a novel method called RD-HRL. Built based on HRL, RD-HRL introduces a reliability-driven decision mechanism, which is composed of an Transition Region Extraction (TRE) module, Target Identification (TI) module, and a Target Evaluation (TE) module, as it is illustrated in Figure 2. The TRE module extracts transition regions from offline trajectories, serving as the candidate spaces where critical decision-level targets are likely to reside. TI selects a decision-level target from these candidate regions, which acts as a reliable intermediate target to facilitate the high-level policy in generating action-level targets. TE evaluates and refines the selected decision-level targets, enabling the optimization of their selection and ensuring reliable guidance for the overall decision-making process. Finally, the high-level policy provides generalization-noise-immune action-level targets for the low-level policy to take actions. In the following, we will first give the details of TRE, TI, and TE of the reliability-driven decision mechanism. Subsequently, we will discuss how to incorporate the reliability-driven decision mechanism with HRL to enhance the decision-making in long-horizon sparse-reward tasks. The pseudo code of RD-HRL can be found in Appendix B.

### 3.1 COMPONENTS OF RELIABILITY-DRIVEN DECISION MECHANISM

In this section, we describe the three essential novel components of our reliability-driven decision mechanism: the **Transition Region Extraction (TRE)** module, the **Target Identification (TI)** module, and the **Target Evaluation (TE)** module.

The **Transition Region Extraction (TRE)** module is responsible for extracting transition regions from the offline dataset, thereby supplying the TI module with the candidate set for decision-level target. Formally, a transition region represents a set of all states within a block, where the block corresponds to the part where two or more trajectories are much closer to each other, for instance, $z$ in Figure 1 (a), $z_1$, $z_2$ and $z_3$ in Figure 2 (a). We form the TRE module with a simple yet effective strategy: first discretize the dataset into blocks, and then filter out the transition regions from these blocks. Given a dataset $\mathcal{D}$, we first perform K-Means clustering (Sculley, 2010) over the states in $\mathcal{D}$:

$$\mathcal{C} = \text{K-Means}(\{\boldsymbol{s} | \boldsymbol{s} \sim \mathcal{D}\}, N), \tag{5}$$

where $N$ represents the number of clusters, $|\mathcal{C}| = N$ and each $c \in \mathcal{C}$ represents a cluster. For notational simplicity, we denote the cluster ID of state $\boldsymbol{s}_t$ as $c_{\boldsymbol{s}_t}$, $0 \leq c_{\boldsymbol{s}_t} \leq N$.

We then identify transition regions among these blocks. Intuitively, transition regions connect different trajectories. Thus, we propose the Future Diversity Index (FDI) as a metric for quantifying the diversity of reachable futures from a region, thereby identifying transition regions. Formally, the FDI with respect to a cluster $c$ is defined as:

$$\text{FDI}(c) = \frac{|\{c_{\boldsymbol{s}_{t+1}} | \boldsymbol{s}_t \in c \text{ and } \boldsymbol{s}_t \in \boldsymbol{\tau}\}| - 2}{N}, \boldsymbol{\tau} \sim \mathcal{D}. \tag{6}$$

The underlying reason is that, by connecting more trajectories, transition regions typically admit a larger set of possible future directions. Note that we subtract 2 in the numerator because, for any given cluster $c$, there must exist $s_t$ s.t. $c_{s_{t+1}} = c$ and must exist $s_{t'}$ s.t. $c_{s_{t'+1}} \neq c$, where $s_t$ and $s_{t'}$ denote two arbitrary states within cluster $c$, $s_t \neq s_{t'}$. In other words, any cluster necessarily admits at least two possible future clusters: transitioning to other clusters, or remaining in the current one. Further, clusters with higher FDIs are considered as transition regions $\mathcal{Z}$. In our paper, we consider any cluster $c$ with $\text{FDI}(c) > 0$ as one of the transition regions. Formally, we have $\mathcal{Z} = \{c \mid \text{FDI}(c) > 0\}$. In this paper, we determine the optimal number of clusters using Within-Cluster Sum of Squares (WCSS) (Edwards and Cavalli-Sforza, 1965; Brusco and Steinley, 2007; Duong et al., 2013), please refer to Appendix C for details.

The **Target Identification (TI)** module is responsible for selecting $z \in \mathcal{Z}$ as the decision-level target. Without loss of generality, we can model TI as any arbitrary neural network parameterized by $\boldsymbol{\theta}_{TI}$, represented as $\text{TI}_{\boldsymbol{\theta}_{TI}}(\boldsymbol{g}^{TI} | \boldsymbol{s}_t, \boldsymbol{g})$, where $\boldsymbol{s}_t$ is the state at time $t$, $\boldsymbol{g}$ represents the overall task goal, and $\boldsymbol{g}^{TI}$ is the decision-level target. Given the current state $\boldsymbol{s}_t$ and the overall task goal $\boldsymbol{g}$, the TI module selects $z \in \mathcal{Z}$ as the decision-level target $\boldsymbol{g}^{TI}$ for the high-level policy. By providing decision-level target for the high-level policy, the introduction of the TI module naturally restricts the decision space of the high-level policy to regions without generalization demands, thereby providing intermediate guidance for the high-level policy and addressing the generalization noise.

The **Target Evaluation (TE)** module is responsible for providing accurate evaluation of the decision-level targets for the TI module. Represented as $\text{TE}_{\theta_{TE}}(\boldsymbol{s}, \boldsymbol{g})$, the TE module estimates values only for $\boldsymbol{s} \in z, z \in \mathcal{Z}$, rather than for all $\boldsymbol{s} \in \mathcal{D}$. Our TE module performs updates only within the transition regions, abstracting the intermediate fine-grained RL steps into a single macro-step, thereby providing the temporal abstraction of value update. More importantly, by learning solely with respect to $z \sim \mathcal{Z}$, our TE module naturally avoids generalization noise, providing reliable learning guidance for the TI module.

### 3.2 RELIABLE DECISION MAKING WITH RD-HRL

We now can couple the reliability-driven decision mechanism with HRL as RD-HRL for reliability-driven decision-making. Given the current state $\boldsymbol{s}_t$ and the overall task goal $\boldsymbol{g}$, the procedure of RD-HRL is given by:

$$\boldsymbol{g}^{TI} \sim \text{TI}_{\theta_{TI}}(\cdot | \boldsymbol{s}_t, \boldsymbol{g}), \quad \boldsymbol{g}^h \sim \pi_{\theta_h}^h(\cdot | \boldsymbol{s}_t, \boldsymbol{g}^{TI}), \quad \boldsymbol{a}_t \sim \pi_{\theta_l}^l(\cdot | \boldsymbol{s}_t, \boldsymbol{g}^h), \tag{7}$$

where $\boldsymbol{g}^{TI}$ is the decision-level target given by the TI module for high-level policy, $\boldsymbol{g}^h$ is the action-level target given by the high-level policy for low-level policy, and $\boldsymbol{a}_t$ is the action to be taken with respect to the current state $\boldsymbol{s}_t$ and task goal $\boldsymbol{g}$.

Having established a basic understanding of the decision-making process in RD-HRL, we proceed to introduce the training objective of RD-HRL. Given transition regions $\mathcal{Z}$ filtered by the TRE module, we first learn the TE module with transition regions $\mathcal{Z}$. Specifically, we first denote the skeleton of $\boldsymbol{\tau}$ with $\mathcal{Z}$ as $\hat{\boldsymbol{\tau}} = \{..., z_i, z_{i+1}, ...\}$. Similar to previous work Park et al. (2024a), our TE module can be optimized by:

$$\mathcal{L}_{\theta_{TE}} = \mathbb{E}_{\boldsymbol{\tau} \sim \mathcal{D}}[||\text{TE}_{\theta_{TE}}(\boldsymbol{s_{t_1}}, \boldsymbol{g}) - (r_{t_1, t_2} + \gamma^{\text{d}(s_{t_1}, s_{t_2})}\text{TE}_{\theta_{TE}^-}(\boldsymbol{s}_{t_2}, \boldsymbol{g})||_2], \tag{8}$$

where $z_1, z_2 \in \hat{\boldsymbol{\tau}}, \boldsymbol{s}_{t_1} \in z_1, \boldsymbol{s}_{t_2} \in z_2$, $\theta_{TE}^-$ denotes the parameters of the target TE module, and $r_{t_1, t_2}$ is designed as:

$$r_{t_1, t_2} = \begin{cases} 1, & \exists \, \boldsymbol{s} \in [z_1, z_2], ||\boldsymbol{s}_t - \boldsymbol{g}|| \leq \epsilon, \\ 0, & \text{otherwise.} \end{cases} \tag{9}$$

Note that since the transition region $z_i$ spans across trajectories, $\boldsymbol{s}_{t_1} \, \boldsymbol{s}_{t_2}$ could be in different trajectories. For the sake of simplicity, we assume $\boldsymbol{s}_{t_1} \in \boldsymbol{\tau}$, there must $\exists \, \boldsymbol{s}_{t_2'} \in \boldsymbol{z_2}, \boldsymbol{s}_{t_2'} \in \boldsymbol{\tau}$ and $\boldsymbol{s}_{t_2'} \sim \boldsymbol{s}_{t_2}$. Therefore, $\text{d}(\boldsymbol{s}_{t_1}, \boldsymbol{s}_{t_2})$ can be approximated by $t_2' - t_1$. The advantages of the Trajectory Evaluation (TE) module arise from two aspects. First, states $\boldsymbol{s}_{t_1}$ and $\boldsymbol{s}_{t_2}$ may originate from different trajectories, which enables direct cross-trajectory propagation of value signals rather than generalization, there by overcoming generalization error. Second, by focusing exclusively on transition regions, the TE module requires only a limited number of value updates, thereby mitigating the accumulation of errors incurred in each update.

Given transition regions $\mathcal{Z}$ as well as the TE module, we then provide the learning procedure of the TI module. Similar to previous works (Park et al., 2024a; Osa et al., 2019; Pateria et al., 2021; Bai

Table 1: The average normalized score of different HRL methods on various environments, with $\pm$ denoting the standard deviation. The mean and standard deviation are computed over 50 random seeds. We emphasize in bold scores within 3 percent of the maximum per task ($\geq 0.97*$MAX). The results marked as "-" indicates that the authors of the corresponding work did not provide corresponding results, nor did they provide the corresponding code.

| Datasets | PlanDQ | MSCP | V-ADT | DTAMP | HD-DA | HILP | HILP-Plan | HIQL | DiffuserLite | RD-HRL |
|---|---|---|---|---|---|---|---|---|---|---|
| antmaze-medium-diverse | **93.0 ± 2.6** | 88.9 ± 2.2 | 52.6 ± 1.4 | 88.7 ± 3.7 | 88.7 ± 8.1 | 43.5± 7.6 | 49.2 ± 5.1 | 86.8 ± 4.6 | 87.6 ± 2.0 | **94.6±2.5** |
| antmaze-medium-play | **92.1±1.7** | **91.3 ± 1.3** | 62.2 ± 2.5 | **93.3 ± 0.9** | 85.8 ± 2.4 | 45.6 ± 4.0 | 46.6 ± 10.4 | 84.1 ± 10.8 | 88.8 ± 3.2 | **94.0±1.2** |
| antmaze-large-diverse | 86.0 ± 3.5 | 83.4 ± 3.2 | 36.4 ± 3.6 | 78.0 ± 8.8 | 83.6 ± 5.8 | 46.0 ± 12.7 | 64.5 ± 10.2 | 88.2 ± 5.3 | 75.2 ± 3.5 | **91.3±4.3** |
| antmaze-large-play | 85.3±6.3 | 86.5 ± 1.1 | 16.6 ± 2.9 | 80.0 ± 3.3 | 80.7 ± 6.1 | 49.0 ± 8.8 | 58.8 ± 11.2 | 86.1 ± 7.5 | 69.4 ± 6.5 | **95.3±2.1** |
| antmaze-ultra-diverse | 70.0 ± 4.5 | 55.1 ± 7.3 | - | 59.2 ± 3.1 | 52.2 ± 6.9 | 21.2 ± 11.2 | 59.2 ± 12.7 | 52.9 ± 17.4 | 69.3 ± 2.5 | **81.1±6.3** |
| antmaze-ultra-play | **71.5±3.3** | 36.0 ± 14.3 | - | 49.9 ± 7.1 | 59.1 ± 5.5 | 22.2 ±11.4 | 50.8 ± 9.6 | 39.2 ± 14.8 | 63.7 ± 4.2 | **72.9±5.1** |
| kitchen-partial | **75.0 ± 7.1** | 36.9 ± 3.3 | 46.0 ± 1.6 | 63.4 ± 8.8 | **73.3 ± 1.4** | 63.9 ± 5.7 | 59.7 ± 5.1 | 65.0 ± 9.2 | 71.4 ± 1.2 | 69.6±7.4 |
| kitchen-mixed | 71.7 ± 2.7 | 44.5 ± 5.3 | 46.8 ± 6.3 | **74.4 ± 1.4** | 71.7 ± 2.7 | 55.5 ± 9.5 | 51.9 ± 8.3 | 67.7 ± 6.8 | 64.8 ± 1.8 | **72.9±1.7** |
| CALVIN | 45.0 ± 19.8 | 49.9 ± 11.5 | - | 51.3 ± 2.9 | 44.6 ± 11.7 | 12.1 ± 5.1 | 14.5 ± 2.5 | 43.8 ± 39.5 | 52.1 ± 1.1 | **68.8±9.7** |

et al., 2024), the TI module can be optimized with an AWR-style objective, as follows:

$$\mathcal{L}_{\theta_{TI}} = \mathbb{E}_{z\in\mathcal{Z}, \boldsymbol{s_z}\in z}[\exp(\beta^{\mathrm{d}(\boldsymbol{s_t},\boldsymbol{s_z})} \cdot A^{TI}(\boldsymbol{s_z},\boldsymbol{s_t},\boldsymbol{g})) \cdot \log(\mathrm{TI}_{\boldsymbol{\theta_{TI}}}(\boldsymbol{s_z}|\boldsymbol{s_t},\boldsymbol{g}))], \qquad (10)$$

where $z$ is a transition region sampled from $\mathcal{Z}$, $\boldsymbol{s_z}$ is a state sampled from $z$ within the same trajectory as $\boldsymbol{s_t}$, $A^{TI}$ represents the advantage function, which can be formulated as $\mathrm{TE}_{\theta_{TE}}(\boldsymbol{s_z},\boldsymbol{g}) - \mathrm{TE}_{\theta_{TE}}(\boldsymbol{s_t},\boldsymbol{g})$. $\mathrm{d}(\boldsymbol{s_t},\boldsymbol{s_z})$ represents the temporal distance between $\boldsymbol{s_t}$ and $\boldsymbol{s_z}$. Different from learning the TE module, for the TI module, we carefully select $\boldsymbol{s_z} \in z$ from the same trajectory as $\boldsymbol{s_t}$ to avoid decision-level uncertainty. Therefore, supposing $\boldsymbol{s_t} \in \boldsymbol{\tau_{s_t}}$, we have $\exists \; \boldsymbol{s_{t'}} \in \boldsymbol{\tau_{s_t}} \; s.t. \; \boldsymbol{s_{t'}} = \boldsymbol{s_z}$; then, we have $\mathrm{d}(\boldsymbol{s_t},\boldsymbol{s_z}) = t' - t$. Note that we introduced a weight raised to the power of $\mathrm{d}(\boldsymbol{s_t},\boldsymbol{s_z})$ in $\mathcal{L}_{\theta_m^\pi}$, as the distance between $\boldsymbol{s_t}$ and $\boldsymbol{s_z}$ ($i.e.$, $t' - t$) is not always 1.

With respect to the conventional HRL components, we first optimize the value function $V_{\theta_V}(\boldsymbol{s},\boldsymbol{g})$. Specifically, with a trajectory $\boldsymbol{\tau}$ being represented as $\boldsymbol{\tau} = \{(\boldsymbol{s_t},\boldsymbol{a_t},r_t,\boldsymbol{s_{t+1}})\}_{0 \leq t \leq T}$, the value function $V$ can be optimized by:

$$\mathcal{L}_{\theta_V} = \mathbb{E}_{\tau\sim\mathcal{D}}[||V_{\theta_V}(\boldsymbol{s_t},\boldsymbol{g}) - (r_t + \gamma V_{\theta_V^-}(\boldsymbol{s_{t+1}},\boldsymbol{g}))||_2], \qquad (11)$$

where $\gamma$ is the discount factor, $r_t = 1_{||\boldsymbol{s_{t+1}}-\boldsymbol{g}||\leq\epsilon}$, $\theta_V^-$ denotes the parameters of the target value function Park et al. (2024a); Florence et al. (2022).

After learning the value function, following previous works (Qing et al., 2024; Wang et al., 2023; Eysenbach et al., 2023; Park et al., 2024a; Osa et al., 2019; Pateria et al., 2021; Bai et al., 2024), the high-level policy $\pi^h_{\theta_h}(\boldsymbol{s_{t+H}}|\boldsymbol{s_t},\boldsymbol{s_z})$ and the low-level policy $\pi^l_{\theta_l}(\boldsymbol{a_t}|\boldsymbol{s_t},\boldsymbol{s_{t+H}})$ are learned with Equation (3) and Equation (4), respectively. Note that for the learning of $\pi^h_{\theta_h}(\boldsymbol{s_{t+H}}|\boldsymbol{s_t},\boldsymbol{s_z})$, we replace $\boldsymbol{g}$ with $\boldsymbol{s_z} \in z, z \in \mathcal{Z}$, as we are now conditioning on transition regions rather than $\boldsymbol{g}$.

## 4 EXPERIMENT

In this section, we conduct experiments to evaluate the performance of RD-HRL and analyze the impact of its key design. The computational resources and hyper-parameters details are available in Appendix F.

### 4.1 EXPERIMENTAL SETUP

**Benchmarks.** We evaluate RD-HRL on long-horizon goal-conditioned tasks and robotic manipulation tasks, which serve as canonical benchmarks for long-horizon sparse-reward problems (Liu et al., 2022). For long-horizon goal-conditioned tasks, we adopt conventional antmaze-{medium, large}-{diverse, play} (Todorov et al., 2012; Brockman et al., 2016), as well as antmaze-ultra-{diverse, play} (Zhang et al.), which feature expanded and more complex maze layouts (requiring extended temporal reasoning with increased spatial challenges) as our benchmarks. For robotic manipulation tasks, we adopt Kitchen (Gupta et al., 2020) and CALVIN (Mees et al., 2022) as our benchmarks.

**Baselines.** We take methods that adopt states $H$-steps further as action-level targets, such as SOTA methods HIQL (Park et al., 2024a), PlanDQ (Chen et al., 2024a), MSCP (Wu et al., 2024), V-ADT (Ma et al., 2023), DTAMP (Hong et al., 2023), HD-DA (Chen et al., 2024b). We also take the methods with median-based action-level target selection methods as baselines, such as HILP (Park et al., 2024b) and HILP-Plan (Park et al., 2024b). Besides, we also take the method that uses three-layer hierarchical design, DiffuserLite (Dong et al., 2024), as one of the baselines.

Table 2: Ablation results of RD-HRL-TRE, RD-HRL-HP, RD-HRL-TE, RD-HRL-CU and RD-HRL. We mark the best results of each task with **bold**.

| Datasets | HIQL | RD-HRL-TRE | RD-HRL-HP | RD-HRL-TE | RD-HRL-CU | RD-HRL |
|---|---|---|---|---|---|---|
| antmaze-medium-diverse | $86.8 \pm 4.6$ | $85.3 \pm 2.2$ | $59.1 \pm 11.9$ | $90.7 \pm 1.9$ | $92.8 \pm 3.1$ | **94.6$\pm$2.5** |
| antmaze-medium-play | $84.1 \pm 10.8$ | $85.8 \pm 5.2$ | $66.7 \pm 7.3$ | $86.8 \pm 3.2$ | $91.4 \pm 2.7$ | **94.0$\pm$1.2** |
| antmaze-large-diverse | $88.2 \pm 5.3$ | $89.2 \pm 2.6$ | $59.0 \pm 8.1$ | $87.9 \pm 4.2$ | $89.4 \pm 1.3$ | **91.3$\pm$4.3** |
| antmaze-large-play | $86.1 \pm 7.5$ | $84.8 \pm 4.9$ | $54.5 \pm 7.3$ | $85.8 \pm 1.3$ | $90.2 \pm 4.3$ | **95.3$\pm$2.1** |
| antmaze-ultra-diverse | $52.9 \pm 17.4$ | $59.8 \pm 7.4$ | $27.8 \pm 3.3$ | $35.3 \pm 4.4$ | $68.1 \pm 2.5$ | **81.1$\pm$6.3** |
| antmaze-ultra-play | $39.2 \pm 14.8$ | $52.9 \pm 9.2$ | $32.6 \pm 9.2$ | $57.8 \pm 4.9$ | $66.0 \pm 2.1$ | **72.9$\pm$5.1** |

## 4.2 OVERALL RESULTS

The results are shown in Table 1. Note that we emphasize in bold scores within 3% of the maximum per task ($\geq 0.97$*MAX), following previous work (Li et al., 2024). Overall, RD-HRL achieves top-3% performance on 8 out of 9 tasks.

With respect to the goal-conditioned tasks, RD-HRL achieves top-3% performance on all of the 6 tasks, demonstrating the effectiveness of RD-HRL on tasks with low-dimensional goal spaces. Especially on antmaze-ultra-{play, diverse}, which are more complex than antmaze-{medium, large}-{play, diverse}, RD-HRL outperforms our backbone method HIQL by 85.9% and 53.3%, demonstrating the superior performance of RD-HRL over conventional HRL methods on tasks with low-dimensional goal spaces.

For manipulation tasks, RD-HRL achieves the top-3% performance across all benchmarks except for the kitchen-partial task. We believe that the poor performance on the kitchen-partial task arises because the dataset lacks trajectories of completion across subtasks, preventing our method from identifying transition regions. Notably, on CALVIN, our method outperforms HIQL by 57%. Given the high dimensionality of the manipulation tasks, these results verify RD-HRL's effectiveness in high-dimensional spaces.

To demonstrate the advantages of RD-HRL more intuitively, we further visualize the decision-level target $g^{TI}$ provided by the TI module on antmaze-ultra-diverse, please refer to Appendix E for details. We additionally evaluate RD-HRL against flat GCRL approaches, please refer to Appendix G.2 for details.

## 4.3 ABLATION STUDY

In this subsection, we conduct further experiments to investigate the key designs of RD-HRL. Specifically, we design the following variants for ablation studies:

- **RD-HRL-TRE**: learning the TI module with state at $t + 2H$ rather than $z \sim \mathcal{Z}$;
- **RD-HRL-HP**: removing high-level planner $\pi^h$ and directly providing $g^{TI}$ to $\pi^l$;
- **RD-HRL-TE**: replacing TE with $V$;
- **RD-HRL-CU**: removing cross-trajectory updating of value from the TE module.

The results are summarized in Table 2. By analyzing the results in Table 2, we can easily conclude the following key findings:

**(1) The benefits of transition regions $\mathcal{Z}$ extracted by the TRE module extend beyond merely enabling larger $H$.** Recall that we propose the Target Identification (TI) module, which provides transition regions extracted by the TRE module as decision-level target $g^{TI}$ for the high-level policy $\pi^h$. In this part, we conduct experiments to validate the effectiveness of learning decision-level target $g^{TI}$ from the transition regions $\mathcal{Z}$.

Specifically, we further propose variant RD-HRL-TRE, which replaces $Z$ as $\{s_{t+2H}\}$, and report the results in Table 2. As can be observed, (1) RD-HRL outperforms RD-HRL-TRE on all of the antmaze tasks, suggesting that the proposed transition region in our method indeed enables reliable action-level target, leading to improved performance. (2) RD-HRL-TRE outperforms HIQL on most tasks, indicating that increasing $H$ may enhance performance to some extent.

**(2) The Target Identification (TI) module goes beyond merely being a higher-level replacement of $\pi^h$.** The Target Identification (TI) module delivers decision-level target to the high-level policy.

One might curious whether TI merely stands as a higher-level replacement of the high-level policy, and whether it would instead be possible to pass such target directly to the low-level policy?

We believe the answers to the aforementioned questions are negative. We answer these via the variant RD-HRL-HP, in which we remove the high-level policy, and directly provide the decision-level target to the low-level policy. The results are summarized in Table 2. Comparison between RD-HRL-HP with RD-HRL reveals a significant performance drop. This is particularly evident in the antmaze-ultra-{diverse, play} environments, where performance declines by 65.2% and 55.3%, respectively. We attribute this to that the Target Identification (TI) module goes beyond being just a higher-level replacement of $\pi^h$. The decision-level target $\boldsymbol{g}^{TI}$ provided by the TI module may be unreachable for low-level policy; in such cases, the high-level policy is required to decompose $\boldsymbol{g}^{TI}$.

**(3) The advantages of Trajectory Evaluation (TE) Module stem both form the direct cross-trajectory value update and the temporal abstraction.** As an ablation, we first propose RD-HRL-TE by replacing the TE module with $V$ in RD-HRL to examine its contribution. As is shown in Table 2, RD-HRL-TE exhibits a certain performance degradation, which is more pronounced in complex environments such as antmaze-ultra-{diverse, play}. This demonstrates that the TE module can handle long-horizon scenarios better than the original value function $V$. Moreover, despite the performance decline, we can see that RD-HRL-TE still achieves results comparable to or better than HIQL in 5 out of the 6 environments, further highlighting the advantages of the TI module introduced by RD-HRL.

We believe the advantages of the TE module stem from the two critical components: the deterministic cross-trajectory value update and temporal abstraction. To assess the contribution of these components, we introduce a variant termed RD-HRL-CU, in which the deterministic cross-trajectory value update is ablated while temporal abstraction is preserved. The corresponding results are reported in Table 2. As can be observed, on the one hand, RD-HRL-CU exhibits a certain degree of performance degradation compared with RD-HRL, with a notable drop of 16.1% on antmaze-ultra-diverse. This directly demonstrates the advantage conferred by the directly cross-trajectory value update. On the other hand, RD-HRL-CU still consistently outperforms RD-HRL-TE across all datasets, providing empirical evidence that the retained temporal abstraction contributes substantially to its effectiveness. We attribute this to the fact that temporal abstraction reduces the frequency of value updates, thereby mitigating the cumulative errors incurred by stepwise updates. Please refer to Appendix D for theoretical proof.

### 4.4 FURTHER INVESTIGATIONS

#### 4.4.1 COMPARISON BETWEEN HORIZON ENLARGEMENT AND RD-HRL

RD-HRL adopts the Target Identification (TI) module to provide decision-level target $\boldsymbol{g}^{TI}$ for high-level policy $\pi^h$. To some extent, this may be regarded as equivalent to extending $H$, since the decision-level target $\boldsymbol{g}^{TI}$ is typically more long-term than high-level targets $\boldsymbol{g}^h$. Moreover, discussions in Section 4.3 indicate that increasing $H$ may enhance performance. Thus, a question arises: **Is the improved performance due to the larger $H$?**

To verify this conjecture, we tested the performance of HIQL under different $H$ and summarized the results alongside RD-HRL's performance in Figure 3. As can be observed, both on antmaze-ultra-play and antmaze-ultra-diverse, with the waysteps $H$ increasing, the performance of HIQL improves, reaching its peak at $H = 50$. In other words, increasing $H$ can improve the performance of HIQL at early stages. Yet, note that our RD-HRL still outperforms HIQL under $H = 50$. After $H = 50$, the performance decreases as $H$ gets larger. We attribute this phenomenon to the fact that as $H$ continues to increase, the action-level targets $\boldsymbol{g}^h$ of HIQL become increasingly difficult for $\boldsymbol{s}_t$ to achieve, consequently leading to performance degradation. As a brief recap, although increasing $H$ does improve the performance, the TI module we introduced is fundamentally what gives RD-HRL its competitive advantage.

#### 4.4.2 EXTENDING TRANSITION REGIONS BEYOND EUCLIDEAN SETTINGS

While the results in Table 2 already demonstrate the feasibility of RD-HRL, specifically the TRE module, in Euclidean-space tasks, many robotic manipulation tasks involve non-Euclidean state spaces, which are harder to tackle. To verify the generality of our method across these distinct

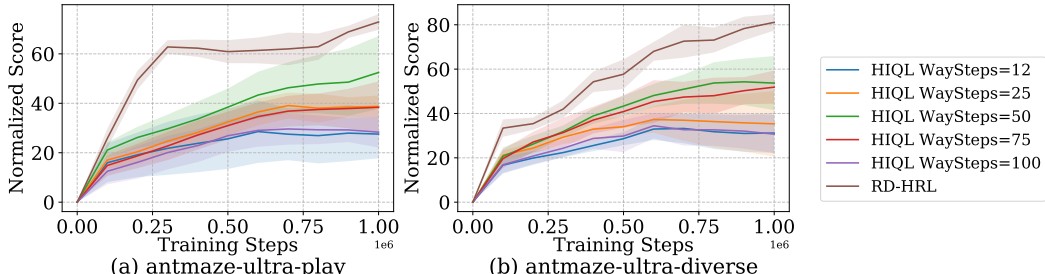

Figure 3: On antmaze-ultra-{play, diverse} environments, RD-HRL's performance is compared with HIQL using different waysteps $H$. While the choice of waysteps $H$ affects HIQL's performance, RD-HRL consistently maintains its leading position.

geometrical settings, we further compare RD-HRL and RD-HRL-TRE on kitchen-partial, kitchen-mixed, and CALVIN. The corresponding results are shown in Table 3.

It can be observed that in non-Euclidean robotic manipulation tasks, directly using $s_{t+2H}$ as a decision-level target yields only limited performance gains over RD-HRL. This highlights the critical role of the transition region in such tasks. Moreover, the performance degradation of RD-HRL-TRE on certain benchmarks (e.g., CALVIN) further substantiates the necessity and effectiveness of the TRE module.

Table 3: Comparison of RD-HRL and RD-HRL-TRE on manipulation tasks.

| Datasets | RD-HRL-TRE | HIQL | RD-HRL |
|---|---|---|---|
| kitchen-partial | $67.6 \pm 3.2$ | $65.0 \pm 9.2$ | $\mathbf{69.6 \pm 7.4}$ |
| kitchen-mixed | $69.1 \pm 5.4$ | $67.7 \pm 6.8$ | $\mathbf{72.9 \pm 1.7}$ |
| CALVIN | $35.5 \pm 2.0$ | $43.8 \pm 39.5$ | $\mathbf{68.8 \pm 9.7}$ |

Therefore, we conclude that extracting the transition region remains feasible and beneficial in non-Euclidean tasks.

### 4.5 SUPPLEMENTARY EXPERIMENTS AND ANALYSIS

Beyond the aforementioned experiments, we performed supplementary studies, such as decision process visualizations and hyperparameter analyses, to validate the effectiveness of RD-HRL. Please refer to Appendix G for details.

## 5 RELATED WORKS

RD-HRL belongs to the narrow class of hierarchical reinforcement learning (HRL) methods. In a broader sense, it is also related to option-based RL methods.

**The narrow class of HRL methods** are commonly designed as a two-level structure where a high-level planner selects sub-goals over a fixed horizon for low-level planners (Pateria et al., 2021; Barto and Mahadevan, 2003; Botvinick, 2012), such as PlanQD (Chen et al., 2024a), HIRO (Nachum et al., 2018) and HD (Chen et al.). Such a commonly adopted design alleviates the credit assignment problem, while making sub-goal selection coarse and rigid. To address this, G-ADT Ma et al. (2024) selects states with the highest attainable value rather than the expected value to promote exploration; DTAMP (Hong et al., 2023) generates the shortest sub-goal paths to improve planning efficiency; RIS (Chane-Sane et al., 2021) and HILPs (Park et al., 2024b) choose intermediate states from value functions and Hilbert space (Young, 1988) representations, respectively. It is worth noting that although some methods adopt fixed-level hierarchical designs with three or more layers (DiffuserLite (Dong et al., 2024) or HAC (Levy et al., 2019) for example), all these methods still rely on noisy value functions for sub-goal selection, resulting in unreliable sub-goals.

**Option-based methods** exploit task semantics to decompose an overall task into semantically meaningful sub-tasks. For example, DDO (Fox et al., 2017) employs a policy-gradient method under behaviour cloning to discover reusable segments (i.e., sub-tasks) from datasets, where each segment can further contain finer-grained ones. Clustering-based approaches, such as Evans and Şimşek (2023), represent the dataset as a graph and apply Louvain clustering (Blondel et al., 2008) for discretisations, treating sub-clusters as low-level action candidates and higher-level clusters as high-level policies, thereby achieving layer-by-layer decomposition without predefining the number of options. Similarly, FraCOs (Cannon and Şimşek, 2025) identifies frequently recurring sub-

sequences within successful trajectories as callable options to build multi-level skill hierarchies, while (Hu et al., 2022) improves the efficiency of hierarchical discovery through causal graphs of environmental variables. Despite their ability to achieve semantical decompositions, option-based HRL methods often incur substantial training and planning costs, which motivates the development of a low-cost HRL approach with reliable sub-goals.

More generally, RD-HRL can be viewed as related to trajectory-stitching methods, while distinguishing itself by providing reliable action-level targets through a reliability-driven decision mechanism. Please refer to Appendix H and J for detailed discussion.

## 6 CONCLUSION AND DISCUSSION

In this paper, we analyze the shortcomings of existing HRL methods when dealing with long-horizon sparse-reward tasks: they select action-level targets ($i.e.$, sub-goals) based on conventional value functions that affected by generalization noise, which leads to sub-optimal trajectories. As the solution, we propose RD-HRL, which introduces the novel reliability-driven decision mechanism to select decision-level target from transition regions for high-level policy, thereby restricting the decision space of high-level policy to local regions without generalization requirements, yielding reliable action-level targets. Theoretical analysis and experimental results demonstrate the superior performance of our method RD-HRL.

**Limitations and Future Directions.** Although our work has demonstrated promising performance on long-horizon sparse-reward tasks, the current learning process for transition regions $\mathcal{Z}$ remains relatively naive, relying exclusively on clustering. The decoupled execution of $\mathcal{Z}$ extraction and policy learning inhibits flexible end-to-end joint optimization of regions and policies, thereby constraining the method's overall performance, especially in high-dimensional observational tasks ($e.g.$, vision-based tasks). Developing an end-to-end method based on RD-HRL could further extend its applicability, representing a crucial direction for future enhancements.

## 7 ETHICS STATEMENT

This research was conducted in accordance with established ethical standards for scientific work. Topics considered include, but are not limited to, the involvement of human subjects, dataset usage and release practices, potentially harmful insights, research methodologies and applications, conflicts of interest and sponsorship, discrimination/bias/fairness concerns, privacy and security issues, legal compliance, and research integrity (e.g., IRB approvals, documentation, and research ethics). Specifically, our study does not involve human subjects or personally identifiable information, and therefore no Institutional Review Board (IRB) approval was required. All datasets used are publicly available and released under appropriate licenses. Potential risks, including fairness, bias, privacy, and unintended harmful use of the findings, were carefully assessed, and steps were taken to minimize such risks. We affirm that our work complies with research integrity guidelines, including accurate reporting, transparency, and reproducibility.

## 8 REPRODUCIBILITY STATEMENT

We have taken multiple steps to ensure the reproducibility of our results. The main text provides detailed descriptions of the model architecture and training procedure, while the appendix includes additional explanations of implementation details and hyper-parameters. All datasets used in our experiments are publicly available. Furthermore, we release the source code, configuration files, execution environment, and pre-processing scripts in an anonymous repository, enabling researchers to faithfully reproduce our experiments.

ACKNOWLEDGMENTS

This work was supported by the National Natural Science Foundation of China (Grants No.62307020, No.U2341229), the National Key R&D Program of China (Grant No.2023YFF0905400), the Fundamental and Interdisciplinary Disciplines Breakthrough Plan of the Ministry of Education of China (Grant No.JYB2025XDXM903), and the New Cornerstone Science Foundation through the XPLORER PRIZE.

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

## A    THE USE OF LARGE LANGUAGE MODELS (LLMS)

We employ Large Language Models (LLMs) for grammar checking in our paper.

## B    PSEUDO CODE OF RD-HRL

---
**Algorithm 1** Training
---
1:  Extract transition regions $\mathcal{Z}$ with the TRE module.
2:  Initialize $V_{\theta_V}$, $\text{TE}_{\theta_{TE}}$, $\text{TI}_{\theta_{TI}}$, $\pi^h_{\theta_h}$, $\pi^l_{\theta_l}$.
3:  **while** not converged **do**
4:      # Learning of $\text{TI}_{\theta_{TI}}$, $\pi^h_{\theta_h}$, $\pi^l_{\theta_l}$.
5:      Update $\theta_{TI}$ with Equation (10).
6:      Update $\theta_h$ with Equation (3).
7:      Update $\theta_l$ with Equation (4).
8:      # Learning of $V_{\theta_V}$, $\text{TE}_{\theta_{TE}}$.
9:      Update $\theta_V$ with Equation (11).
10:     Update $\theta_{TE}$ with Equation (8).
11: **end while**
---

---
**Algorithm 2** Evaluating
---
1:  Initialize environment *env*, obtain current state $s_t$ and goal $g$.
2:  done $\leftarrow$ False.
3:  **while** not done **do**
4:      # Obtain decision-level target $g^{TI}$ with $\text{TI}_{\theta_{TI}}(g^{TI}|s_t, g)$.
5:      $g^{TI}_t \leftarrow \text{TI}_{\theta_{TI}}(\cdot|s_t, g)$.
6:      # Obtain action-level target $g^h$ with $\pi^h_{\theta_h}(g^h|s_t, g^{TI})$.
7:      $g^h_t \leftarrow \pi^h_{\theta_h}(\cdot|s_t, g^{TI}_t)$.
8:      # Obtain action $a$ with $\pi^l_{\theta_l}(a_t|s_t, g^h)$.
9:      $a_t \leftarrow \pi^l_{\theta_l}(\cdot|s_t, g^h_t)$.
10:     # Interact with environment *env*.
11:     new state $s_{t+1}$, reward, done $\leftarrow env(a_t)$.
12: **end while**
---

## C    DETERMINE OF NUMBER OF CLUSTERS $N$

Many studies have proposed pioneering methods for determining the optimal number of clusters in K-Means (Kodinariya et al., 2013; Shi et al., 2021; Zhang et al., 2020). In this paper, we determine the optimal number of clusters by computing the inflection point of the Within-Cluster Sum of Squares (WCSS) (Edwards and Cavalli-Sforza, 1965; Brusco and Steinley, 2007; Duong et al., 2013) of various numbers of clusters, in which WCSS is defined as:

$$\text{WCSS} = \sum_{C_i \in C} \sum_{s \in C_i} ||s - \mu_i||^2, \tag{12}$$

where $C$ is the set of clusters and $C_i \in C$. WCSS calculates the sum of squared distances from all samples to their respective cluster centroids; a smaller WCSS value indicates that the samples are closer to their cluster centers, signifying better clustering performance. Consequently, the optimal cluster number can be conveniently selected by evaluating WCSS values across varying cluster numbers.

Specifically, for each dataset, we define its candidate set for the number of clusters $N$ as $\{N_1, N_2, ..., N_n\}$, where $N_1 < N_2 < ... < N_n$. For a fixed dataset, as $N$ increases, the decrease in WCSS is inevitable because each cluster becomes smaller, thereby reducing the distance between samples and their cluster centroids. However, this process is not linear. When the number of clusters $N$ is smaller than the optimal number of clusters $\hat{N}$, WCSS decreases more rapidly; whereas

when $N$ exceeds $\hat{N}$, the rate of WCSS reduction slows down. Consequently, the optimal number of clusters $\hat{N}$ can be determined by detecting the point where the WCSS reduction rate exhibits a notable transition. In our paper, we select $N$ as $\hat{N}$ which has the maximum third-order derivative, as it represents where the WCSS trend exhibits a significant change.

## D  THEORETICAL PROOF FOR THE ADVANTAGE OF THE TARGET EVALUATION MODULE

**Proposition 1.** *The Target Evaluation module yields lower accumulated noise.*

*Proof of Proposition 1.* Under long-horizon sparse-reward settings, the credit assignment issue tends to propagate noise in value function updates. Let us hypothesize that, for the ground-truth value $V(\boldsymbol{s}_t)$, every update incorporates a Gaussian noise scaled by $\beta$. Assuming the value function after $N$-step updates is $V(\boldsymbol{s}_{t-N})$, we can now quantitatively analyze the noise introduced by $N$-step updates by calculating the entropy of $V(\boldsymbol{s}_{t-N})$ and $V(\boldsymbol{s}_t)$. In other words, $h(V(\boldsymbol{s}_{t-N})) - h(V(\boldsymbol{s}_t))$ quantifies the amount of noise introduced. For ease of presentation, we omit the condition $\boldsymbol{g}$ and parameters $\theta$ in this proof.

First of all, we need the analytical form of $V(\boldsymbol{s}_{t-N})$. For a state $\boldsymbol{s}_t$, we have:

$$V(\boldsymbol{s}_t) = \gamma V(\boldsymbol{s}_{t+1}) + r_t + \beta V(\boldsymbol{s}_{t+1})\epsilon_t, \tag{13}$$

where $\epsilon_t$ is an independent Gaussian noise, $\epsilon \sim N(0,1)$. As $\gamma$ and $\beta$ are fixed parameters, we denote $\gamma = \alpha\beta$. Then, we have $V(\boldsymbol{s}_{t-N})$ for $N \geq 2$:

$$V(\boldsymbol{s}_{t-N}) = \gamma V(\boldsymbol{s}_{t-N+1}) + r_{t-N} + \beta V(\boldsymbol{s}_{t-N+1})\epsilon_{t-N} \tag{14}$$

$$= \alpha\beta V(\boldsymbol{s}_{t-N+1}) + r_{t-N} + \beta V(\boldsymbol{s}_{t-N+1})\epsilon_{t-N} \tag{15}$$

$$= (\alpha\beta + \beta\epsilon_{t-N}) \cdot V(\boldsymbol{s}_{t-N+1}) + r_{t-N} \tag{16}$$

$$= (\alpha\beta + \beta\epsilon_{t-N})[(\alpha\beta + \beta\epsilon_{t-N+1}) \cdot V(\boldsymbol{s}_{t-N+2}) + r_{t-N+1}] + r_{t-N} \tag{17}$$

$$= (\alpha\beta + \beta\epsilon_{t-N})(\alpha\beta + \beta\epsilon_{t-N+1}) \cdot V(\boldsymbol{s}_{t-N+2}) + (\alpha\beta + \beta\epsilon_{t-N}) \cdot r_{t-N+1} + r_{t-N} \tag{18}$$

$$= (\alpha\beta + \beta\epsilon_{t-N})(\alpha\beta + \beta\epsilon_{t-N+1})(\alpha\beta + \beta\epsilon_{t-N+2}) \cdot V(\boldsymbol{s}_{t-N+3}) \tag{19}$$

$$+ (\alpha\beta + \beta\epsilon_{t-N+1}) \cdot r_{t_N+2}(\alpha\beta + \beta\epsilon_{t-N})(\alpha\beta + \beta\epsilon_{t-N+1}) \cdot r_{t-N+1} \tag{20}$$

$$+ r_{t-N} \tag{21}$$

$$= \ldots \tag{22}$$

$$= V(\boldsymbol{s}_t)\Pi_{k=0}^{N-1}(\alpha\beta + \beta \cdot \epsilon_{t-N+k}) \tag{23}$$

$$+ \sum_{j=1}^{N-1}[r_{t-j} \cdot \Pi_{k=0}^{N-j-1}(\alpha\beta + \beta\epsilon_{t-N+k})] \tag{24}$$

$$+ r_{t-N} \tag{25}$$

$$\approx V(\boldsymbol{s}_t)\Pi_{k=0}^{N-1}(\alpha\beta + \beta \cdot \epsilon_{t-N+k}). \tag{26}$$

Note that as we are considering the long-horizon sparse-reward settings, we consider $r = 0$ for simplicity.

The Gaussian process with multiplicative noise leads to computational complexity. For simplicity, since the logarithmic transformation $log(\cdot)$ is injective, which does not affect the differential entropy of the distribution, given Equation (26), we have:

$$log(V(\boldsymbol{s}_{t-N})) = log(V(\boldsymbol{s}_t)) + \sum_{k=0}^{N-1} log(\alpha\beta + \beta \cdot \epsilon_{t-N+k}) \tag{27}$$

$$= log(V(\boldsymbol{s}_t)) + \mathcal{N}((N-1)\alpha\beta, (N-1)\beta^2). \tag{28}$$

Assuming $V(\boldsymbol{s}_t) = \mathcal{N}(\mu_t, \sigma_t^2)$, we have $V(\boldsymbol{s}_{t-N}) = \mathcal{N}(\mu_t + (N-1)\alpha\beta, \sigma_t^2 + (N-1)\beta^2)$. Then we have the differential entropy of $log(V(\boldsymbol{s}_t))$ and $log(V(\boldsymbol{s}_{t+N}))$ as:

$$h(log(V(\boldsymbol{s}_t))) = \frac{1}{2}log(2\pi e\sigma_t^2) + log(\mu_t) - \frac{1}{2}(\frac{\sigma_t}{\mu_t})^2, \tag{29}$$

$$h(log(V(\boldsymbol{s}_{t+N}))) = \frac{1}{2}log(2\pi e(\sigma_t^2 + (N-1)\beta^2)) + log(\mu_t + (N-1)\alpha\beta) - \frac{1}{2}\frac{\sigma_t^2 + (N-1)\beta^2}{(\mu_t + (N-1)\alpha\beta)^2}. \tag{30}$$

With Equation (29) and Equation (30), we have the information of noise introduced by the plain value function $V$:

$$\eta_V = h(log(V(\boldsymbol{s}_{t+N}))) - h(log(V(\boldsymbol{s}_t))). \tag{31}$$

Meanwhile, with our temporal abstracted value function TE, the steps required to update from $\text{TE}(\boldsymbol{s}_t)$ to $\text{TE}(\boldsymbol{s}_{t+N})$ required $N' \ll N$ steps, as the temporal abstracted value function TE is learned on the *sketleon* of trajectories. Although, similarly, we have the information of noise introduced by our temporal abstracted module TE as:

$$\eta_{TE} = h(log(\text{TE}(\boldsymbol{s}_{t+N}))) - h(log(V(\boldsymbol{s}_t))), \tag{32}$$

where $h(log(\text{TE}(\boldsymbol{s}_{t+N})))$ is the differential entropy of logged updated temporal abstracted module TE. Then we have:

$$\eta_{\text{TE}} - \eta_V = [h(log(\text{TE}(\boldsymbol{s}_{t+N}))) - h(log(V(\boldsymbol{s}_t)))] - [h(log(V(\boldsymbol{s}_{t+N}))) - h(log(V(\boldsymbol{s}_t)))] \tag{33}$$

$$= h(log(\text{TE}(\boldsymbol{s}_{t+N}))) - h(log(V(\boldsymbol{s}_{t+N}))) \tag{34}$$

$$= \frac{1}{2}log(2\pi e(\sigma_t^2 + (N'-1)\beta^2)) + log(\mu_t + (N'-1)\alpha\beta) - \frac{1}{2}\frac{\sigma_t^2 + (N'-1)\beta^2}{(\mu_t + (N'-1)\alpha\beta)^2} \tag{35}$$

$$- (\frac{1}{2}log(2\pi e(\sigma_t^2 + (N-1)\beta^2)) + log(\mu_t + (N-1)\alpha\beta) - \frac{1}{2}\frac{\sigma_t^2 + (N-1)\beta^2}{(\mu_t + (N-1)\alpha\beta)^2}) \tag{36}$$

$$= \frac{1}{2}log(2\pi e(\sigma_t^2 + (N'-1)\beta^2)) - \frac{1}{2}log(2\pi e(\sigma_t^2 + (N-1)\beta^2)) \tag{37}$$

$$+ log(\mu_t + (N'-1)\alpha\beta) - log(\mu_t + (N-1)\alpha\beta) \tag{38}$$

$$+ \frac{1}{2}\frac{\sigma_t^2}{(\mu_t + (N-1)\alpha\beta)^2} - \frac{1}{2}\frac{\sigma_t^2}{(\mu_t + (N'-1)\alpha\beta)^2} \tag{39}$$

$$+ \frac{1}{2}\frac{(N-1)\beta^2}{(\mu_t + (N-1)\alpha\beta)^2} - \frac{1}{2}\frac{(N'-1)\beta^2}{(\mu_t + (N'-1)\alpha\beta)^2}. \tag{40}$$

As $N' \ll N$, we have:

$$\frac{1}{2}log(2\pi e(\sigma_t^2 + (N'-1)\beta^2)) - \frac{1}{2}log(2\pi e(\sigma_t^2 + (N-1)\beta^2)) < 0, \tag{41}$$

$$log(\mu_t + (N'-1)\alpha\beta) - log(\mu_t + (N-1)\alpha\beta) < 0, \tag{42}$$

$$\frac{1}{2}\frac{\sigma_t^2}{(\mu_t + (N-1)\alpha\beta)^2} - \frac{1}{2}\frac{\sigma_t^2}{(\mu_t + (N'-1)\alpha\beta)^2} < 0. \tag{43}$$

For the last factor, we define

$$f(N) = \frac{N-1}{(\mu_t + (N-1)\alpha\beta)^2}, N \geq 0. \tag{44}$$

Then we have:

$$\frac{\partial f}{\partial N} = \frac{(\mu_t + (N-1)\alpha\beta)^2 - 2\alpha\beta(\mu_t + (N-1)\alpha\beta)(N-1)}{(\mu_t + (N-1)\alpha\beta)^4} \tag{45}$$

$$= \frac{(\mu_t + (N-1)\alpha\beta) - 2\alpha\beta(N-1)}{(\mu_t + (N-1)\alpha\beta)^3} \tag{46}$$

$$= \frac{\mu_t - \alpha\beta(N-1)}{(\mu_t + (N-1)\alpha\beta)^3}. \tag{47}$$

Note that $\gamma = \alpha\beta$ is set close to 1 in RL, and under the sparse reward setting, we have $0 \leq \mu_t \ll N$, thus $\frac{\partial f}{\partial N} < 0$, $f(N)$ is monotonically decreasing. Further, with $N' < N$, we have:

$$\frac{1}{2}\frac{(N-1)\beta^2}{(\mu_t + (N-1)\alpha\beta)^2} - \frac{1}{2}\frac{(N'-1)\beta^2}{(\mu_t + (N'-1)\alpha\beta)^2} = \frac{\beta^2}{2}(f(N) - f(N')) < 0. \tag{48}$$

Considering Equation (40), Equation (41), Equation (42), Equation (43) and Equation (48) together, we now have $\eta_{\text{TE}} - \eta_V < 0$. In conclusion, the Target Evaluation (TE) module yields lower noise.

$\square$

## E    ANALYSIS OF DECISION-LEVEL AND ACTION-LEVEL TARGET PREDICTION

To better illustrate the Target Identification (TI) module in RD-HRL, taking antmaze-ultra-diverse as an example, we show how the decision-level target steers the agent toward the final target in Figure 6, and show how the high- and low-level policies accomplish the decision-level target in Figure 5. We have also visualised trajectories in the dataset, and the filtered transition regions $\mathcal{Z}$ of antmaze-ultra-diverse in Figure 4. As can be observed in Figure 6, the TI module generates decision-level targets $g^{TI}$ from transition regions $\mathcal{Z}$, guiding the high- and low-level policy towards the task goal. Diving into Figure 5, we can observe that the agent achieves an action-level target under the guidance of high-level policy; after the agent achieves the decision-level target, the TI module produces new decision-level targets.

## F    EXPERIMENTAL ENVIRONMENT AND HYPERPARAMETERS

RD-HRL is trained using Flax under JAX (Sapunov, 2023) on an Ubuntu 22.04 LTS server, with $4 \times$ NVIDIA A40 (Ampere architecture, 48GB VRAM each), 72-core processor (dual-socket Intel Xeon Platinum), and 503GB memory.

We design the TI module, TE module, value function, high-level and low-level policy as MLPs. The hyper-parameters are summarized in Table 4, please refer to Table 4 for details.

## G    ADDITIONAL EXPERIMENTAL RESULTS

### G.1    PRELIMINARY ATTEMPT IN VISUAL SCENARIOS

RD-HRL has demonstrated significant advantages in goal-conditioned tasks and robotic manipulation tasks. Nevertheless, scaling RD-HRL to visual scenarios is a promising idea.

We believe this can be solved in two ways: (1) As we discussed in Section 6, we could try to design the pipeline as an end-to-end process. In this scenario, filtering transition regions could be performed on task-relevant embeddings generated by a jointly-learned encoder. Or, more generally, (2) we could attempt to use a general large models to understand high-dimensional observations and generate universal embeddings, then perform clustering on these embeddings. However, these

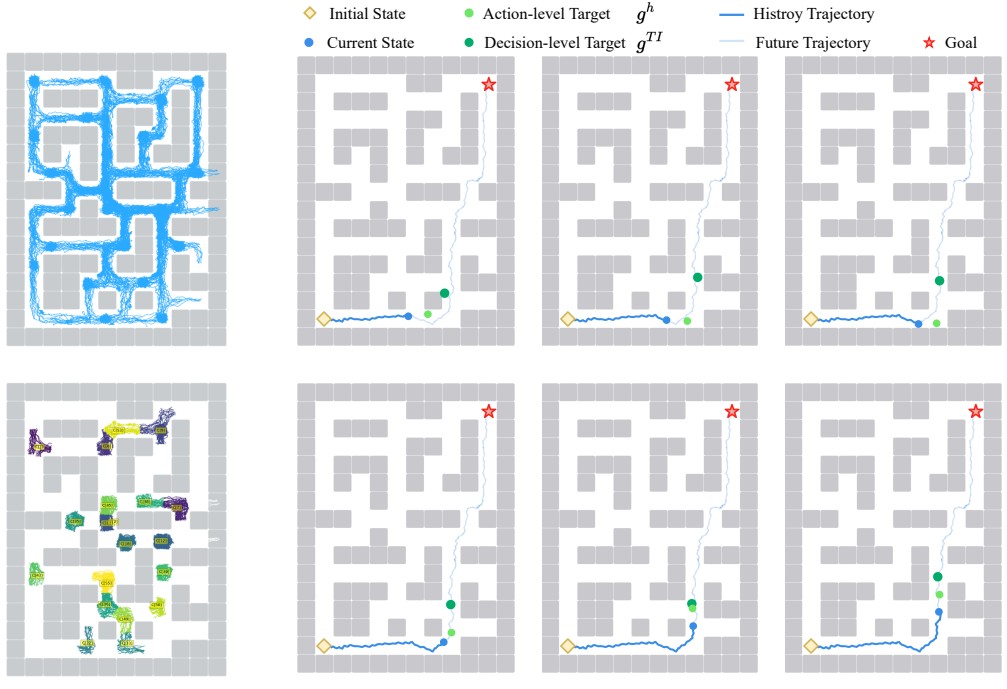

Figure 4: Trajectories in dataset (upper) and extracted transition regions (lower).

Figure 5: After the low-level policy achieved the decision-level target, the TI module produces decision-level targets by sampling from further transition regions.

Table 4: Detail of Hyper-parameters. We set the other hyperparameters to be consistent with those in HIQL Park et al. (2024a).

| Hyperparameter | Value |
|---|---|
| Value discounts | 0.99 |
| Goal dimensions | 10 (kitchen, CALVIN), 29 (Antmaze) |
| Training steps | 1000000 |
| Batch size | 1024 |
| TI module dimensions | (256, 256) |
| TI module dimensions | (512, 512, 512) |
| Policy MLP dimensions | (256, 256) |
| Value MLP dimensions | (512, 512, 512) |
| Representation MLP dimensions | (512, 512, 512) |
| Activation | GELU |
| Optimizer | Adam |
| Learning rate | 0.0003 |
| Target network decay rate | 0.005 |

embeddings might be task-agnostic, and this approach suffers from the drawbacks of separate execution discussed in Section 6. Therefore, we believe solution (1) is a more promising direction. We will leave solution (1) as our future work.

Nevertheless, in this section, we present an preliminary attempt to apply RD-HRL to more challenging visual scenarios. Specifically, we proposed the following variants and evaluated their performance on a visual environment, procgen-500 (Park et al., 2024a):

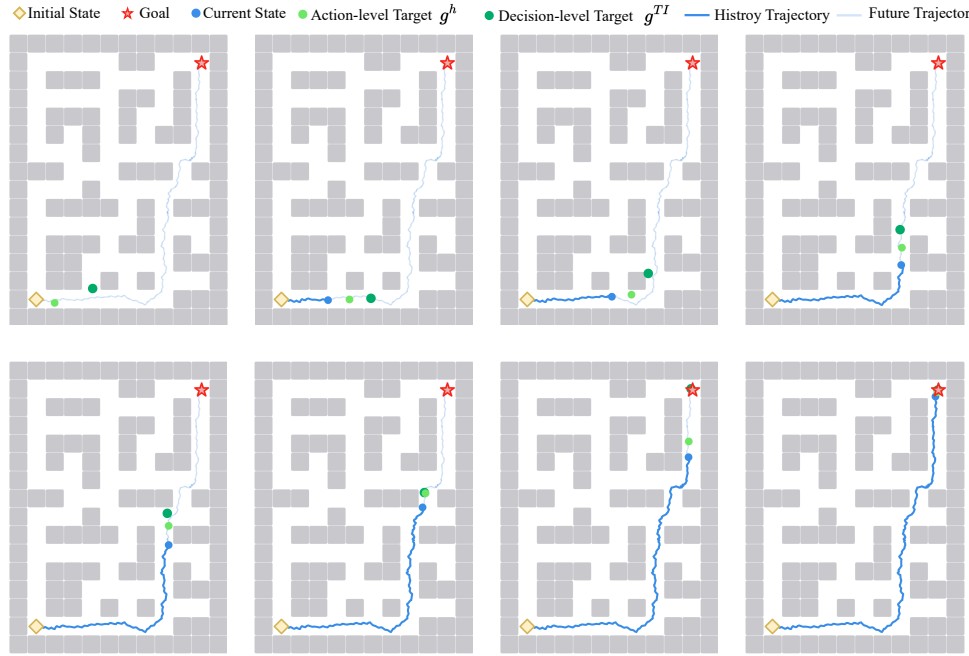

Figure 6: By generating decision-level targets, TI module coordinates the high-level and low-level planning processes.

- RD-HRL$^e$: Uses embeddings from DINOv2 (Oquab et al.) for discretization and filtering of trajectory-adjacent regions. DINOv2 is a pretrained Vision Transformer (ViT) (Han et al., 2022) model. The results of RD-HRL$^e$ serve to validate the performance of RD-HRL in a general embedding scenario.

- RD-HRL$^{e+}$: Uses embeddings from HIQL for discretization and filtering of transition regions. As a HRL method, the encoder of HIQL produces task-relevant embedding; therefore, the results of RD-HRL$^{e+}$ preliminarily verify the performance of RD-HRL in a task-relevant embedding scenario. Note that we use PCA (Maćkiewicz and Ratajczak, 1993) to reduce the size of HIQL's embedding to match that of DINOv2's embedding before performing transition regions extraction.

- RD-HRL: After flattening the raw RGB observations, we apply PCA to reduce their size to match that of DINOv2's embeddings before performing transition regions extraction.

The results are summarized in Table 5. Note that Performance (train) refers to the performance on the environment levels used for training the policy, whereas Performance (test) refers to the performance on the environment levels used for testing the policy.

Table 5: Results of RD-HRL$^e$, RD-HRL$^{e+}$ and RD-HRL on visual task procgen-500.

| Methods | RD-HRL | RD-HRL$^e$ | RD-HRL$^{e+}$ |
|---|---|---|---|
| **Performance (train)** | $3.5 \pm 2.2$ | $11.5 \pm 5.1$ | $13 \pm 3.9$ |
| **Performance (test)** | 0 | $2 \pm 1.7$ | $7.5 \pm 4.3$ |

It can be observed that both at the train level and the test level, when the transition region extraction is performed directly on the observation space, it is hard for RD-HRL to complete the tasks. However, when the transition region extraction is performed directly on the general embeddings (RD-HRL$^e$), RD-HRL achieves a certain success rate, indicating that general embeddings can help alleviate the limitations of RD-HRL in high-dimensional observation scenarios. Moreover, RD-HRL$^{e+}$ achieves

Table 6: The average normalized score of different GCRL methods on various environments, with ± denoting the standard deviation. The mean and standard deviation are computed over 50 random seeds. We emphasize the best scores of each task in **bold**.

| Datasets | GCBC | GC-POR | GC-IQL | RD-HRL |
|---|---|---|---|---|
| antmaze-medium-diverse | 67.3±10.1 | 74.8 ± 11.9 | 63.5 ± 14.6 | **94.6±2.5** |
| antmaze-medium-play | 71.9±16.2 | 71.4 ± 10.9 | 70.9 ± 11.2 | **94.0±1.2** |
| antmaze-large-diverse | 20.2±9.1 | 49.0 ± 17.2 | 50.7 ± 18.1 | **91.3±4.3** |
| antmaze-large-play | 23.1±15.6 | 63.2 ± 16.1 | 56.5 ± 14.4 | **95.3±2.1** |
| antmaze-ultra-diverse | 14.4±9.7 | 29.8 ± 13.6 | 21.6 ± 15.2 | **81.1±6.3** |
| antmaze-ultra-play | 20.7±9.7 | 31.0 ± 19.4 | 29.8 ± 13.6 | **72.9±5.1** |
| kitchen-partial | 38.5±11.8 | 18.4 ± 14.3 | 39.2 ± 13.5 | **69.6±7.4** |
| kitchen-mixed | 46.7±20.1 | 27.9 ± 17.9 | 51.3 ± 12.8 | **72.9±1.7** |
| CALVIN | 17.3±14.8 | 12.4 ± 18.6 | 7.8 ± 17.6 | **68.8±9.7** |

better results than RD-HRL[e], demonstrating that task-relevant embeddings can further address this limitation.

In summary, although this is only a rough initial validation, the results offer preliminary support for our future work aimed at enhancing RD-HRL's capability in high-dimensional settings, demonstrating the feasibility of our future work.

### G.2 COMPARISON OF RD-HRL WITH FLAT GCRL METHODS

We additionally evaluate RD-HRL against conventional GCRL approaches, including GCBC (Ding et al., 2019), GC-POR (Xu et al., 2022), and GC-IQL (Kostrikov et al.; Park et al., 2024a). The results are summarized in Table 6. As can be observed, RD-HRL outperforms the baselines on all of the 9 tasks, demonstrating the advantages of RD-HRL.

### G.3 HOW ABOUT APPLYING THE TARGET EVALUATION (TE) MODULE ON HIGH-LEVEL PLANNER, INSTEAD OF THE ADDITIONAL TI MODULE?

We introduce a TI module trained with a Target Evaluation (TE) module that provides low-noise value estimates for decision-level target selection, yielding promising results. However, such a design raises a natural question: **Can the TE module be applied directly to the high-level planner in a standard two-level HRL architecture, and obviate the TI module?**

Before addressing this question further, we would like to clarify a crucial point: the TE module and TI module are complementary components that operate synergistically, the TE module is fundamentally designed to estimate the value of transition regions, not for any states in the dataset. However, the high-level planner requires $V(\boldsymbol{s}, \boldsymbol{g})$ for any $\boldsymbol{s} \in \mathcal{S}$ and any $\boldsymbol{g} \in \mathcal{G}$, which is a capability that exceeds the scope of the TE module.

Nevertheless, we conducted experiments and propose variant RD-HRL-TI, which directly applies the TE module to the high-level policy $\pi^h$. Results are summarized in Table 7. As can be observed, compared to HIQL, directly applying the TE module to $\pi^h$ brings only a negligible performance improvement, which is far inferior to applying the TE module to the TI module. This suggests that combining the TE module with the TI module is a more reasonable design.

Table 7: Comparison among HIQL, RD-HRL and RD-HRL-TI.

| Datasets | HIQL | RD-HRL-TI | RD-HRL |
|---|---|---|---|
| antmaze-ultra-play | 39.2 ± 14.8 | 40.0 ± 8.9 | **72.9 ± 5.1** |
| antmaze-ultra-diverse | 52.9 ± 17.4 | 54.7 ± 4.2 | **81.1 ± 6.3** |

### G.4    HOW SENSITIVE IS RD-HRL TO THE NUMBER OF CLUSTERS?

To further explore the impact of cluster number $N$ on RD-HRL, we evaluate RD-HRL with various values of $N$. The results are summarized in Table 8. As shown, the performance improved as the number of clusters increased, peaking at N=60. Further increasing the number of clusters led to a performance decline. We believe this is because: (a) With fewer clusters, the divisions of clusters were too rough, preventing the TI module from accurately learning the accurate decision-level target. (b) As the number of clusters grew, their size decreased. This resulted in more clusters having high FDI, which obscured the advantage of true transition regions over other regions.

Table 8: Performance of RD-HRL with various $N$.

| Datasets | N = 10 | N = 20 | N = 40 | N = 60 | N = 80 | N = 100 |
|---|---|---|---|---|---|---|
| antmaze-ultra-play | $53.3 \pm 7.3$ | $52.8 \pm 2.2$ | $66.2 \pm 3.9$ | $\mathbf{72.9 \pm 5.1}$ | $59.1 \pm 1.2$ | $55.7 \pm 2.8$ |
| antmaze-ultra-diverse | $52.0 \pm 9.2$ | $69.2 \pm 1.1$ | $77.0 \pm 2.9$ | $\mathbf{81.1 \pm 6.3}$ | $73.1 \pm 2.6$ | $59.7 \pm 3.3$ |

### G.5    IS THE TRAINING OVERHEAD INCURRED BY THE TI MODULE AND THE TE MODULE JUSTIFIABLE?

To evaluate the acceptability of these additional computational costs introduced by the TI module and the TE module, we measured the training time consumption of HIQL and RD-HRL across different datasets, as is summarised in Table 9. It can be observed that the introduction of the TI module and the TE module indeed brings about a decline in training efficiency; however, even for the slowest learning speed of 82 iterations per second with a batch size of 32 on antmaze-ultra-play, RD-HRL only requires 3.4 hours to learn $10^6$ steps, meaning RD-HRL remains efficient.

In summary, although the introduction of the TI module and the TE module increases computational overhead, the overall execution time of RD-HRL remains within an acceptable range, thanks to the efficiency of the JAX framework.

Table 9: Comparison of training effiency of HIQL and RD-HRL.

| Datasets | HIQL (iterations/s) | RD-HRL (iterations/s) | RD-HRL (1M steps/h) |
|---|---|---|---|
| antmaze-ultra-play | 257 | 82 | 3.4 |
| antmaze-ultra-diverse | 261 | 84 | 3.3 |
| antmaze-large-play | 249 | 87 | 3.2 |
| antmaze-large-diverse | 253 | 79 | 3.5 |
| antmaze-medium-play | 255 | 88 | 3.1 |
| antmaze-medium-diverse | 262 | 83 | 3.3 |
| kitchen-mixed | 311 | 129 | 2.1 |
| kitchen-partial | 315 | 132 | 2.1 |
| CALVIN | 326 | 125 | 2.2 |

### G.6    INVESTIGATIONS OF ALTERNATIVES OF K-MEANS

To explore clustering methods beyond K-Means (Ahmed et al., 2020), we experimented with the density-based method DBSCAN (Deng, 2020) and the graph-augmented method HDB-SCAN (McInnes et al., 2017), and visualized their clustering results in Figure 7. As is shown, the clustering results of both DBSCAN and HDBSCAN turned out to be chaotic, making it impossible for us to infer transition regions or conduct further experiments based on those cluster methods. We believe this is because that DBSCAN and HDBSCAN assume that clusters consist of density-connected regions, but trajectory data typically form sparse and non-uniform structures, making density-based assumptions unreliable.

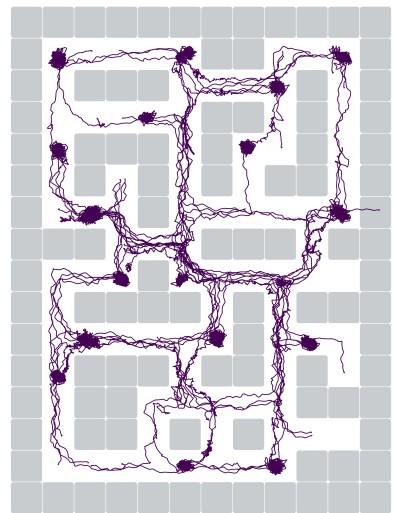 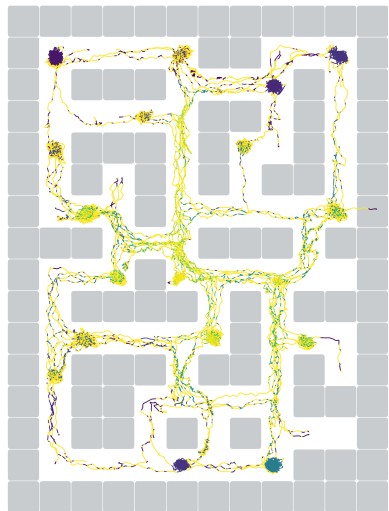

(a) DBSCAN                                        (b) HDBSCAN

Figure 7: Cluster results of antmaze-ultra-diverse with DBSCAN and HDBSCAN.

Table 10: Comparison of HGCBC and RD-HGCBC.

| Dataset | HGCBC | RD-HGCBC |
|---|---|---|
| antmaze-ultra-diverse | 39.4 | **42.0** |
| antmaze-ultra-play | 38.2 | **49.8** |

### G.7    APPLICATION OF RD-HRL ON HGCBC

To verify whether the reliability-driven decision mechanism proposed in RD-HRL can also benefit other methods, we incorporate the reliability-driven decision mechanism into HGCBC and propose RD-HGCBC. The results are summarized in Table 10. As shown, RD-HGCBC achieves clear improvements on both antmaze-ultra-play and antmaze-ultra-diverse, with a particularly notable gain of 30.4% on antmaze-ultra-play. This demonstrates the applicability of the RD framework to other algorithms.

### G.8    THE IMPACT OF THE FDI THRESHOLDING OF THE PERFORMANCE OF RD-HRL.

We have also evaluated how different FDI thresholding choices affect model performance, and we summarize the results in Table 11. As can be observed, as the FDI threshold becomes more permissive, the model's performance gradually degrades. The drop is particularly pronounced when the FDI × N decreases from 3 to 2. This is because most non–transition regions have two future clusters; thus, when we relax the FDI threshold from 3 to 2, the model is exposed to a large number of noisy transition regions, which leads to the observed performance degradation.

Table 11: The impact of the FDI thresholding.

| FDI * N + 2 | 1 | 2 | 3 |
|---|---|---|---|
| antmaze-ultra-play | 56.6 | 60.3 | **72.9** |
| antmaze-ultra-diverse | 62 | 64.3 | **81.1** |

### G.9    ANALYSIS OF TRANSITION REGIONS IN MANIPULATION TASKS

We further visualize the transition regions extracted from the kitchen-mixed environment in Figure 8 to demonstrate the soundness of RD-HRL in manipulation tasks. As is shown, the frames in which the robot arm is positioned between the table and the cabinet are marked as transition regions. This is because the trajectories in kitchen-mixed follow a consistent ordering: the agent typically manipulates the kettle and microwave on the table first, then moves on to the light switch and burner bottom in the middle, and finally opens or slides the cabinet doors at the top. Consequently, the intermediate frames, where the arm is located between the table and the cabinet, serves as a state that connects tasks on the table (kettle and microwave) with tasks on the top (light switch, burner bottom or the cabinet doors). This illustrates that our method effectively identifies transition regions in manipulation tasks. Furthermore, the results in Table 1 provide evidence of our method's performance on manipulation benchmarks.

### G.10    ANALYSIS OF SELECTED TRANSITION REGIONS AND DOWNSTREAM TASK SUCCESS

We computed the probability distribution over the transition regions selected by TI across 3,500 trajectories, and we also measured the normalized score associated with each selected transition region. The results are summarized in Figure 9, in which the the horizontal axis represents the IDs of the 18 extracted transition regions, labeled sequentially from 0 to 17; the vertical axis shows the probability of being selected after passing through a given transition region and the average normalized score obtained after passing through that transition region, respectively.

It can be observed that the transition regions chosen by the TI module correspond to high success rates, demonstrating that the TI module's selection of transition regions is positively correlated with downstream task success. In addition, transition region 14 provides a representative example of a low-return transition region. Its extremely low selection probability indicates that our method is capable of avoiding transition regions that yield low success rates for the current state. In other words, TI's selection of transition regions is closely related to the success of downstream tasks.

### G.11    IMPACT OF FDI ON TRANSITION REGIONS EXTRACTION

Theoretically, by connecting more trajectories, transition regions typically admit a larger set of possible future directions. To further illustrate the role of FDI in identifying transition regions, we visualize the transition regions selected under different FDI values as is shown in Figure 10. In summary, using $FDI \times N > 0$ (i.e., $FDI > 0$) as the selection criterion yields more reasonable transition regions.

### G.12    EXTENDING RD-HRL TO ONLINE REINFORCEMENT LEARNING

We have also transferred the RD-HRL to HAC (Levy et al., 2019) and propose RD-HAC. The results are summarized in Figure 11. As shown, in the early and middle stages (700k - 1500k steps), RD-HAC exhibits a significant advantage. (During the first 0–700k steps, the replay buffer is still sparsely populated. As a result, the RD mechanism cannot yet fully demonstrate its advantages.) As exploration progresses (after 1500k steps), the performance of RD-HAC and HAC becomes similar. However, RD-HAC achieves the best success rate of 91% at 1200k steps, whereas HAC reaches its best success rate of 89% at 1900k steps, demonstrating that the RD mechanism still shows its advantage in the online setting.

We believe this stems from RD's high-level understanding of the environment and its efficient use of samples. In the early and middle stages, the replay buffer collected by HAC during exploration is insufficient to support a thorough understanding of the environment, leading to the possibility of suboptimal decisions. In contrast, in RD-HAC, by extracting transition regions, the introduced RD mechanism enables it to have a comprehensive understanding of the environment even with a limited number of samples. As a result, RD-HAC outperforms HAC in the early and middle stages. However, as the samples collected by HAC during exploration become more abundant, the performance of HAC should improve faster, making the advantages brought by RD may no longer be as pronounced.

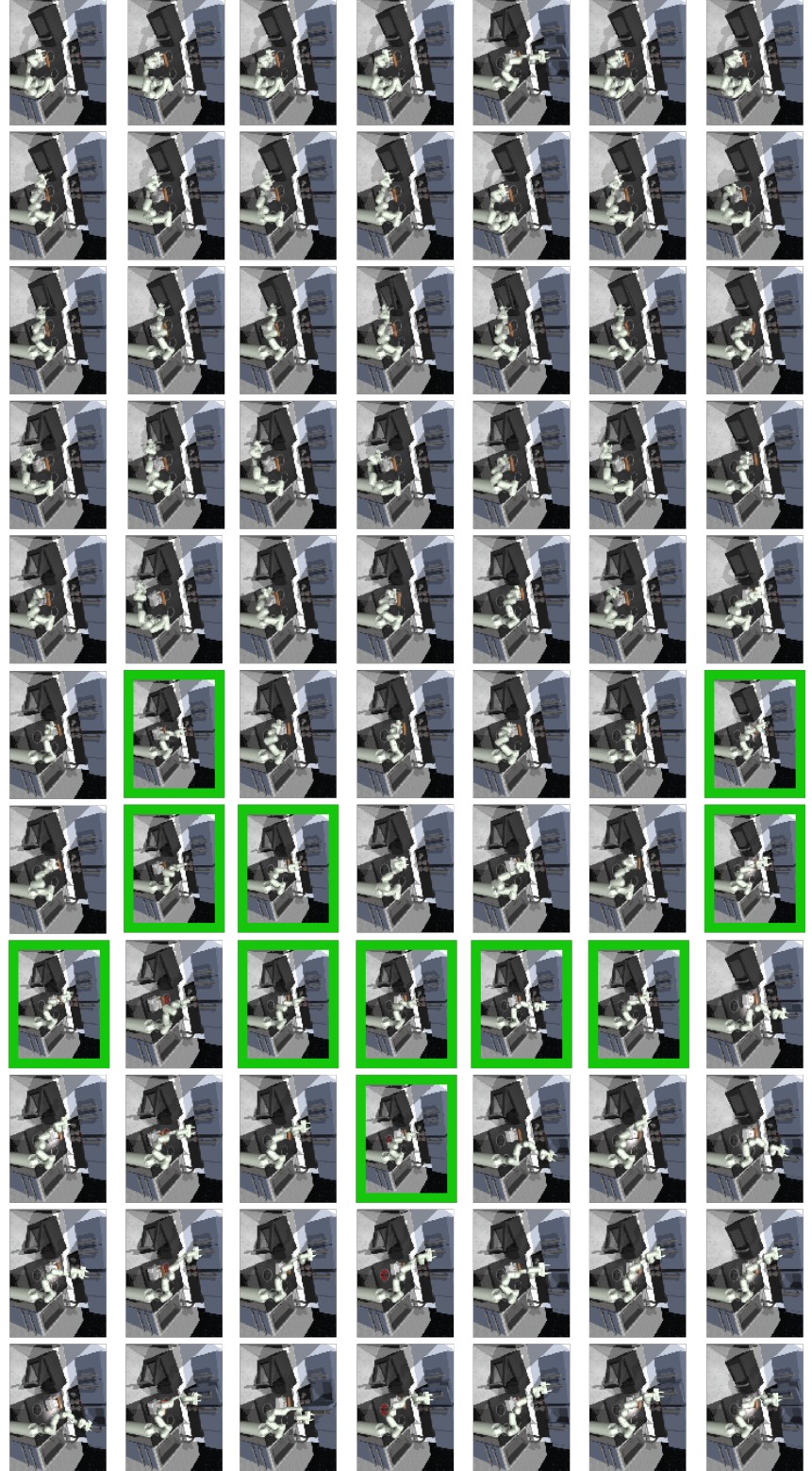

Figure 8: Example of transition regions in kitchen-mixed.

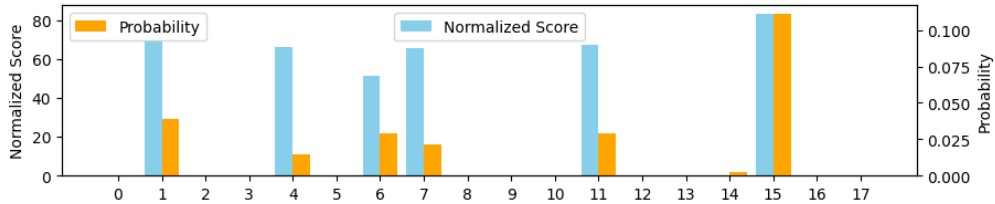

Figure 9: Probability of transition regions and normalized scores achieved after each transition region in antmaze-ultra-diverse.

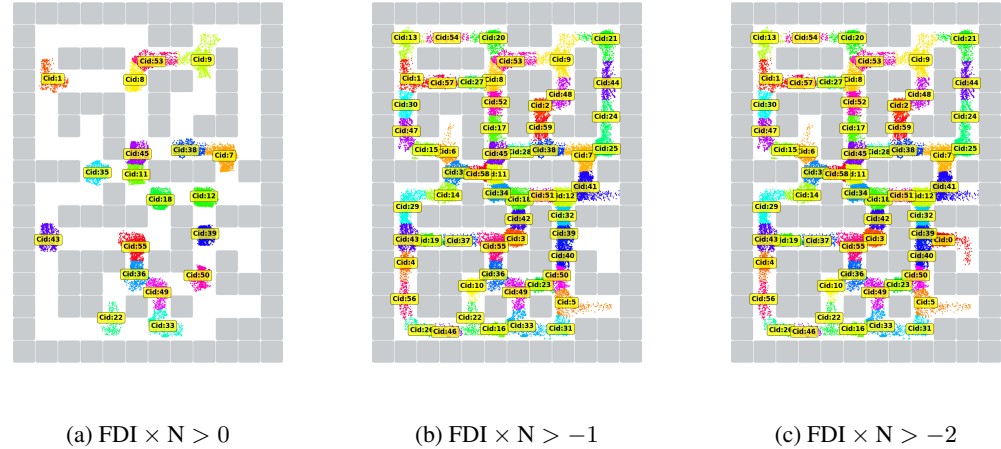

(a) FDI × N > 0          (b) FDI × N > −1          (c) FDI × N > −2

Figure 10: transition regions extracted with different FDI thresholds.

## H  THE STITCHING CAPABILITY OF RD-HRL

Prior work has suggested that an optimal value function can naturally achieve trajectory stitching, as it implicitly organizes the dataset into a graph structure (Char et al., 2022). However, this optimality assumption is often unrealistic, especially in long-horizon sparse-rewards tasks (Wen et al., 2024; Kazemnejad et al., 2024). Consequently, the cumulative noise and generalization noise of flat value functions hinder existing HRL methods from performing reliable policy-level trajectory stitching.

Fortunately, by introducing both the TI module and the TE module, RD-HRL provides a principled guarantee of trajectory stitching. On the one hand, the TI module constructs a decision domain for

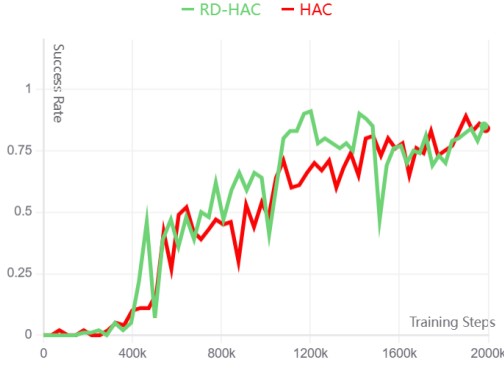

Figure 11: Comparison of HAC and RD-HAC on ant-four-rooms.

the high-level and low-level policies that is free from accumulated errors and generalization noise. Within the local decision domain partitioned by the decision-level targets, the conventional value function remains optimal for the high-level and low-level policies. On the other hand, the TE module provides the TI module with a decision basis that is likewise robust against accumulated errors and generalization noise. Overall, the design of RD-HRL enables each policy to make decisions guided by (locally) optimal value functions, thereby ensuring reliable policy-level stitching.

## I    DISCUSSION OF UNRELIABLE SUB-GOALS

A large body of work can serve as indirect support for unreliable sub-goals. Take Figure 1 as an example, unreliable sub-goals arise from unreliable value functions, which in turn stem from the value functions' reliance on generalization. This introduces indirect supervision signals for $s_{t+H}^1$ in effect. For instance, in pseudo-labeling approaches (Pham et al., 2021; Rizve et al., 2021; Li et al., 2023), pseudo-labels are regarded as generalized supervisory signals, and numerous studies employ consistency regularization to reinforce the dominance of direct supervision in model learning (Abuduweili et al., 2021; Wu et al., 2022), or use thresholding of pseudo-labels to directly discard low-confidence generalized signals (Guo and Li, 2022). This implies that the signals from direct samples are more reliable than those from indirect samples (pseudo labels).

## J    DISCUSSION OF RD-HRL AND OTHER STITCHING METHODS

As is discussed in Appendix H, our method is also analogous to trajectory stitching methods in a broader sense. Here, we provide a detailed analysis of trajectory stitching methods and discuss in detail the differences and connections between RD-HRL and these methods.

To better leverage offline data and approach the dataset's optimal policy, many works focus on trajectory stitching. Some methods operate at the dataset level for data augmentation. For example, BATS (Char et al., 2022) trains a tabular MDP on collected data and uses learned dynamics models to generate short connecting trajectories, which are then used to train the policy. To overcome the limitations of Bellman completeness, MBRCSL (Zhou et al.) applies dynamic programming to stitch together segments from distinct trajectories without relying on Bellman completeness. Other approaches focus on stitching subsequences to construct higher-quality trajectories, such as TS (Hepburn and Montana, 2022), which searches for candidate next states that lead to higher returns. In addition, Venkatraman et al. enhances multi-modal data modeling by imposing batch constraints on out-of-distribution (OOD) data, combined with diffusion to achieve embedding-level augmentation.

Other methods enable trajectory stitching at the policy level by introducing additional policy designs. Focusing on the stitching ability of value or Q-functions, COG-RL (Singh et al., 2020) employs model-free dynamic programming based on Q-learning to stitch together different skills, while QCS (Kim et al., 2024) builds on this idea by introducing an assistance function as a regularization term to improve Q-guided stitching. With the rise of sequential decision-making approaches, several works have attempted trajectory stitching through the Decision Transformer (DT) (Chen et al., 2021). For example, EDT (Wu et al., 2023) enhances stitching by learning maximum achievable returns. ADT (Ma et al., 2024) further combines a hierarchical learning framework, where higher levels provide value or goal prompts to lower levels to improve stitching. However, ADT still does not explicitly design for stitching, instead relying on the value function for implicit stitching. Similarly, QDT (Yamagata et al., 2023) augments DT's stitching capability by leveraging Q-networks to relabel the return-to-go (RTG).

Our method RD-HRL broadly falls into the policy-level stitching category. Similar to existing fixed-level HRL methods, the stitching ability of policy-level stitching also relies on the value function, which is highly susceptible to the noise introduced by long-horizon sparse-reward setting, as is discussed in Section 1. The noise immunity provided by the TI module and the TE module distinguishes RD-HRL from these methods.

