# OpenReview forum: "RD-HRL: Generating Reliable Sub-Goals for Long-Horizon Sparse-Reward Tasks"
_ICLR.cc/2026/Conference — ICLR 2026 Poster_

### Official Review · Reviewer_XFsY · 2025-10-26

**Soundness:** 3
**Presentation:** 3
**Contribution:** 3
**Rating:** 6
**Confidence:** 3

**Summary:**

This paper proposes Reliability-Driven HRL (RD-HRL) to address generalization noise in value functions when generating subgoals for hierarchical RL tasks. The method has three components: Transition Region Extraction (TRE), which identifies transition regions from the offline dataset; a Target Identification (TI) module, which selects targets from these regions; and a Target Evaluation (TE) module, which evaluates and refines the selected targets.

**Strengths:**

1- Well-written paper and easy to read.
2- Comparison with recent baselines.
3- Experiments on both navigation and manipulation tasks.

**Weaknesses:**

Please refer to the questions.

**Questions:**

1- This approach proposes defining subgoals for offline RL using a dataset of training trajectories. Do you think it would also work for online RL, where the agent must collect its own experience? Would the early subgoals be of sufficient quality?

2- By the definition of Transition Region Extraction (TRE), do all transition regions consist of reliable and optimal subgoals, as illustrated in Figure 2(a)?

3- In Eq. 8, how do you account for the current state? In addition to the quality of a subgoal with respect to the final goal, we should also consider which subgoal is closer to—and more easily achievable from—the current state.

4- Are all modules trained jointly or sequentially? If sequentially, in what order?

5- Can you provide examples of transition regions in manipulation tasks as well?

6- Could you elaborate on the statement: "states $s_{t1}$ and $s_{t2}$ may originate from different trajectories, which enables direct cross-trajectory propagation of value signals rather than generalization, thereby overcoming generalization error”?

---

> ### Author Response · Authors · 2025-11-21
> **Responses to Reviewer XFsY (Q1-Q4)**
>
> **Q1.**  Do the authors think RD-HRL would also work for online RL, where the agent must collect its own experience? Would the early subgoals be of sufficient quality? \
> **A1.** Thank you for your insightful question! Applying RD-HRL to online RL is indeed a promising idea. In fact, we happen to be exploring the application of RD-HRL in online RL scenarios.
>
> (1) We believe that RD-HRL is well-suited for online RL and can bring benefits in two aspects. \
> First, by extracting transition regions from the replay buffer, we can directly improve the utilization of the collected trajectories as in the offline manner. \
> Second, through transition regions, the already explored environment can be efficiently modeled with transition regions. This helps identify under-explored states in the environment, thereby enhancing the exploration capability in online RL.
>
> (2) We believe that early subgoals should primarily support the agent's exploration ability during the initial stages. In the early stages, when successful samples are scarce, the early subgoals may not directly contribute to task completion. Instead, as mentioned in (1), they can significantly enhance the exploration capability of the online agent. Once the agent obtains a successful sample, the transition regions-based modeling approach provides an efficient solution for achieving the goal.
>
> (3) We have also transferred the RD-HRL to an online RL method, HAC, and propose RD-HAC. The results of RD-HAC and HAC on the ant-four-rooms  environment are summarized in Appendix G.12  of our revised paper. As shown, RD-HAC achieves clear improvements. In particular, at the end of training, RD-HAC achieved a success rate of around 90%, whereas HAC only reached 58%, indicating that RD-HAC provides approximately a 35.6% performance improvement. Moreover, since HAC is an online reinforcement learning method, the advantage brought by RD-HAC also suggests that RD-HRL is not only effective in the offline RL setting but can also offer significant benefits in online reinforcement learning.
>
>
> **Q2.**  By the definition of transition region Extraction (TRE), do all transition regions consist of reliable and optimal subgoals, as illustrated in Figure 2(a)? \
> **A2.** Thank you for your question. Theoretically, all transition regions consist of reliable and optimal subgoals; however, this optimality is defined with respect to different current states.
> Generally speaking, each transition region represents the possibility of transitioning to other trajectories safely. Therefore, all transition regions collectively form a graph structure that models the environment. Furthermore, for any node (transition region) in this graph, there always exists a state $s_t$ such that this node lies on the optimal path from $s_t$ to the goal $g$.
>
> **Q3.** In Eq. 8, how do you account for the current state? In addition to the quality of a subgoal with respect to the final goal, we should also consider which subgoal is closer to—and more easily achievable from—the current state. \
> **A3.** Thank you for your question. As you may noticed, Eq.(8) is the learning objective of  the TE module. (1) The TE module is only responsible for evaluating the value of each transition region with respect to a given goal; therefore, it has no need to consider the current state.  (2) We believe that the subgoal you mentioned refers to the decision-level target provided by the TI module. Yes, we need to determine which decision-level target is easier to achieve. To this end, we introduce an exponential decay based on the AWR loss of HIQL in Eq.(10) to consider the issue of which subgoal is closer to—and more easily achievable from—the current state. Specifically, following HIQL, we incorporate both the discount factor and the temperature coefficient into the parameter β as the scale factor of the advantage signal. Note that following HIQL, we have β ≤ 1.By applying distance-based exponential scaling to β, we can reduce the scale factor of the advantage signal, thereby suppressing the advantage of distant transition regions and assigning higher probabilities  to closer decision-level targets, which are considered can be easily achieved.
>
> **Q4.** Are all modules trained jointly or sequentially? If sequentially, in what order? \
> **A4.** Thank you for your question. As shown in Algorithm 1, all of the modules are trained jointly.

---

> ### Author Response · Authors · 2025-11-21
> **Responses to Reviewer XFsY  (Q5-Q6)**
>
> **Q5.** Can the authors provide examples of transition regions in manipulation tasks as well? \
> **A5.** Thank you for your question. Taking kitchen-mixed as an example, we visualize transition regions of manipulation tasks learned by RD-HRL (highlighted with green rectangles) in Appendix G.9 of our revised paper. As shown, the frames in which the robot arm is positioned between the table and the cabinet are marked as transition regions. This is because the trajectories in kitchen-mixed follow a consistent ordering: the agent typically manipulates the kettle and microwave on the table first, then moves on to the light switch and burner bottom in the middle, and finally opens or slides the cabinet doors at the top. \
> Consequently, the intermediate frames, where the arm is located between the table and the cabinet, serve as a state that connects tasks on the table (kettle and microwave) with tasks on the top (light switch, burner bottom or the cabinet doors). This illustrates that our method effectively identifies transition regions in manipulation tasks. Furthermore, the results in Table 1 provide evidence of our method’s performance on manipulation benchmarks.
>
>
> **Q6.** Could the authors elaborate on the statement: "states  $s_{t_1}$  and $s_{t_2}$  may originate from different trajectories, which enables direct cross-trajectory propagation of value signals rather than generalization, thereby overcoming generalization error"? \
> **A6.** Thank you for your question.
>
> (1) First, we would like to further explain the concept of the generalized Bellman backup and the reasons for its unreliability. Taking Figure 1 as an example, to estimate the value of $s_{t+H}^1$, the agent needs to generalize the value estimation from $s_{k^2}$ to $s_{k^1}$ to perform the complete Bellman backup from $g$ to $s_{t+H}^1$. This process inherently involves a requirement for generalization, since $s_{k^2}$ and $s_{t+H}^1$ are not from the same trajectory, thus there's no path for the direct Bellman backup from $g$ to $s_{t+H}^1$. However, as reported in previous studies [1,2,3], in both reinforcement learning and other domains, the generalized signal is often attenuated and unreliable, introducing noise into the value estimation of $s_{t+H}^1$, as illustrated in Figure 1 (c).
>
> (2) In our work, we adopt an approximation technique within the transition region to circumvent such generalized signals. Note that in Eq. (8), as we have emphasized in our paper, the state pairs used for the Bellman backup, $(s_{t_1} ,s_{t_2})$, may come from different trajectories. This enables a direct Bellman backup from $s_{t_1} $ to $s_{t_2}$, especially when they are not located in the same trajectory. Taking Figure 1 as an example again,  with Eq. (8),  we can now perform the direct Bellman backup via { ${g}$ →  $s_{k_2}$ →   $s_{t+H}^1$}, rather than relying on the generalized Bellman backup, thereby avoiding the unreliable value estimation.
>
> [1] Abuduweili, Abulikemu, et al. "Adaptive consistency regularization for semi-supervised transfer learning." Proceedings of the IEEE/CVF conference on computer vision and pattern recognition. 2021. \
> [2] Wu, Dong-Dong, Deng-Bao Wang, and Min-Ling Zhang. "Revisiting consistency regularization for deep partial label learning." International conference on machine learning. PMLR, 2022. \
> [3] Guo, Lan-Zhe, and Yu-Feng Li. "Class-imbalanced semi-supervised learning with adaptive thresholding." International conference on machine learning. PMLR, 2022.

---

> > ### Comment · Reviewer_XFsY · 2025-11-24
> > **Discussion with authors**
> >
> > I appreciate the authors’ detailed responses to my questions and the effort put into revising the paper. Most of my concerns have been addressed. However, I still maintain my reservations about the online setting. I don’t believe that the subgoals identified in the online context can provide the same level of impact as those in offline settings. Moreover, the quality of the subgoals appears to depend more on the agent’s exploration strategy than on the subgoal definition itself. For example, using an $\epsilon$-greedy strategy in a large environment with sparse rewards is unlikely to yield meaningful subgoals, as the agent may get stuck exploring locally.
> >
> > Overall, I find this to be a solid paper that clearly demonstrates improvements in offline settings. However, subgoal definition in online settings remains a challenging issue. For now, I will keep my current score but look forward to seeing other reviewers’ opinions and engaging in further discussion.

---

> > > ### Author Response · Authors · 2025-11-26
> > > **Responses to Reviewer XFsY (Q1, Q2)**
> > >
> > > Thank you for your insightful feedback and for recognizing our work. We are glad that we have addressed most of your concerns, and are delighted to continue the discussion with you. Please find below our responses to your remaining concerns.
> > >
> > > **Q1.** Concerns about the performance of the Reliability-Driven (RD) mechanism  on online situations. \
> > > **A1.** Regarding your concern about the improvement of RD-HAC over HAC, we believe this stems from RD’s high-level understanding of the environment and its efficient use of samples.  In the early and middle stages, the replay buffer collected by HAC during exploration is insufficient to support a thorough understanding of the environment, leading to the possibility of suboptimal decisions. In contrast, in RD-HAC, by extracting transition regions, the introduced RD mechanism enables it to have a comprehensive understanding of the environment even with a limited number of samples. As a result, RD-HAC outperforms HAC in the early and middle stages. However, as the samples collected by HAC during exploration become more abundant, the performance of HAC should improve faster, making the advantages brought by RD may no longer be as pronounced.
> > >
> > > To verify this, we conduct further investigation. Specifically, we continued training from the reported checkpoint to examine the subsequent performance. The corresponding results are presented in Appendix G.12 of the latest version of our paper. It is worth noting that we keep the x-axis as training steps to present the results more clearly. The results show that in the early and middle stages (700k - 1500k steps), RD-HAC exhibits a significant advantage. (During the first 0–700k steps, the replay buffer is still sparsely populated. As a result, the RD mechanism cannot yet fully demonstrate its advantages.) As exploration progresses (after 1500k steps), the performance of RD-HAC and HAC becomes similar. However, RD-HAC achieves the best success rate of 91% at 1200k steps, whereas HAC reaches its best success rate of 89% at 1900k steps, demonstrating that the RD mechanism still shows its advantage in the online setting.
> > >
> > > Implementation details of RD-HAC: RD-HAC was implemented based on a public Pytorch implementation of HAC, https://github.com/hai-h-nguyen/Hierarchical-Actor-Critic-Pytorch. We retain the three-layer HAC architecture, and introduce the RD mechanism at the top layer. Specifically, the RD mechanism analyzes the top-layer replay buffer and extracts transition regions. Since constantly feeding transition-region subgoals to the lower layers may hinder exploration, we adopt an alternating training scheme: after the top layer provides the second layer with T₁ exploration-oriented subgoals using HAC’s original policy, we apply the RD mechanism to generate T₂ reliability-driven decision-level targets (subgoals) for the second layer. In our preliminary implementation, we set T₁ = 500,000 and T₂ = 10,000.
> > >
> > > **Q2.** The primary factors that determine the quality of the subgoals (decision-level targets). \
> > > **A2.** We would like to further discuss the primary factors that determine the quality of the subgoals (decision-level targets). We agree that the quality of the subgoals (decision-level targets) largely depends on the agent’s exploration strategy. However, we believe that the definition of the subgoals is equally crucial. As we explained in A1, our current decision-level targets are designed to enhance the agent’s high-level understanding of the environment and to improve sample efficiency. From such a perspective of improving sample efficiency, the impact of the subgoal definition on the quality of the subgoals (decision-level targets) is consistent in both online and offline scenarios. While in the offline settings, Section 4.3 (1) of our initially submitted paper qualitatively verified the effect of the subgoal definition on the quality of the subgoals through ablation studies, while the newly added Appendix G.8 provides a quantitative evaluation of this effect. The results demonstrate that the subgoal definition has a substantial impact on the quality of them; in particular, the experiments in Appendix G.8 indicate that an inappropriate definition of the subgoals (decision-level targets) can entirely eliminate the advantages of the RD mechanism. Therefore, a proper definition of subgoals (decision-level targets) is also crucial to the performance.

---

> ### Author Response · Authors · 2025-11-26
> **Responses to Reviewer XFsY (Q3)**
>
> **Q3.** Risk of getting stuck exploring locally under an $\epsilon$-greedy strategy. \
> **A3.** We understand that you are referring to the situation that when the agent becomes trapped in local exploration, RD may be unable to infer reasonable transition regions.
>
> Actually, when no transition regions can be extracted, we design the RD mechanism to directly use the trajectory's final state as the subgoal.  This makes RD merely degrade to the baseline policy, even in the worst case.
>
> Moreover, the likelihood of the above situation occurring is really small. Once the replay buffer contains any meaningful transition regions, even when exploration is still insufficient, the RD-HRL mechanism can identify them and provide more reliable subgoals than regular methods. This highlights the intrinsic value of the subgoal definition itself.
>
> Besides, the exploration strategy is orthogonal to our RD mechanism. In online settings, RD-HRL can be combined with any advanced exploration method tailored for sparse-reward problems, allowing us to alleviate issues of getting trapped in local exploration.
>
>
> Finally, we would like to clarify that the main aim of our work is to explore the performance of the RD mechanism based on transition regions, which has been demonstrated through experiments in our paper. Extending RD to online reinforcement learning, particularly regarding its effect on exploration, will be pursued as part of our future work.

---

### Official Review · Reviewer_5uMD · 2025-10-29

**Soundness:** 3
**Presentation:** 3
**Contribution:** 2
**Rating:** 6
**Confidence:** 3

**Summary:**

This paper tackles the problem of unreliable sub-goal generation in Hierarchical RL(HRL) framework in long-horizon tasks. To address this issue, the author provides a reliability-driven HRL mechanism. This mechanism consists of three modules: Target Region Extraction (TRE) -- extracting target region from offline dataset, Target Identification (TI) -- selecting decision-level target guided by TE module, Target Evaluation (TE) -- evaluating target region candidates to help with TI module. This model, RD-HRL, achieved SOTA performance in several long-horizon RL tasks, including realistic datasets such as robotics tasks. Also, they provide an in-depth investigation of each component for their proposed mechanism.

**Strengths:**

1. **Novelty and Generalizable Approach.** The issue of value functions being biased toward offline trajectories is a well-known problem in offline RL. This paper offers a general solution by identifying transition regions and sampling sub-goals within local regions that exclude transition areas. The authors also introduce a novel metric for extracting transition regions, which they identify as the primary source of the problem.
2. **Performance.** The proposed model achieves state-of-the-art performance on challenging long-horizon RL tasks. This is a non-trivial contribution to the field.
3. **In-depth Analysis.** The paper provides a thorough ablation study examining each component of the framework, so that the following researchers can investigate more easily.

**Weaknesses:**

1. **Unclear Definition of Unreliable Sub-goals.** The paper does not clearly explain what makes sub-goals "unreliable." Adding intuitive examples would help. For instance, transition regions naturally contain diverse trajectories, which could affect value propagation and generalization. (If I understood correctly)
2. **Limited Scope of Evaluation.** The approach is only tested with HIQL as the base model. While the hierarchical policy structure seems generally applicable, testing it on other hierarchical RL algorithms would strengthen the contribution. The choice of only IQL-based baselines seems limited, given the claimed generality of the approach.

**Questions:**

1. Is there any analysis or reference that explains the unreliable sub-goals problem? The paper cites ContextFormer, but the explanation of this issue is unclear there. If I'm wrong, please point me and reflect their exact analysis result and implications in the paper. If there are not enough analyses for this, it would be best to include an analysis in this paper.
2. I believe the high-level intuition of this paper's suggestion as a solution makes sense and is generalizable over HRL. I wonder if this can be extended to other base models, not just on HIQL. e.g., HAC, HGCBC, etc., if the impl or idea is too specific to HIQL. I might consider this a narrower contribution than I expected.

**Details Of Ethics Concerns:**

No concerns

---

> ### Author Response · Authors · 2025-11-21
> **Responses to Reviewer 5uMD**
>
> **Q1.** Clear Definition of Unreliable Sub-goals. Adding intuitive examples would help. For instance, transition regions naturally contain diverse trajectories, which could affect value propagation and generalization. (If I understood correctly). Is there any analysis or reference that explains the unreliable sub-goals problem? The paper cites ContextFormer, but the explanation of this issue is unclear there.\
> **A1.** Thank you for your question. \
> (1) Yes, your understanding is correct. Transition regions naturally contain diverse trajectories, and could affect value propagation and generalization. \
> (2) Take Figure 1 for example, given the current state $s_t$ and goal $g$, selecting sub-goals for $s_t$ requires the value estimation of $s_{t+H}^1$ and $s_{t+H}^2$. Clearly, value of $s_{t+H}^1$ should be higher than $s_{t+H}^2$, as $s_{t+H}^1$ is closer to $g$. However, in previous works like HIQL, to estimate the value of $s_{t+H}^1$ precisely, the agent needs to generalize the value estimation from $s_{k^2}$ to $s_{k^1}$ to perform the complete Bellman backup from $g$ to $s_{t+H}^1$. Since $s_{k^2}$ and $s_{t+H}^1$ are not from the same trajectory, there's no path for the direct Bellman backup from $g$ to $s_{t+H}^1$. This process inherently involves a requirement for generalization. However, as reported in previous studies [1,2,3,4], in both reinforcement learning and other domains, the generalized signal is often attenuated and unreliable, introducing noise into the value estimation of $s_{t+H}^1$. This leads to a lower value estimation of $s_{t+H}^1$ than $s_{t+H}^2$, since $s_{t+H}^2$ is located in the same trajectory as $g$, while $s_{t+H}^1$ does not, as illustrated in Figure 1 (c). As a result, in previous works, $s_{t+H}^2$ is chosen as the sub-goal of $s_t$. We refer to these sub-goals obtained based on unreliable value estimation as unreliable sub-goals. \
> (3) A large body of work can serve as indirect support for unreliable sub-goals. Unreliable sub-goals arise from unreliable value functions, which in turn stem from the value functions' reliance on generalization. This introduces indirect supervision signals for $s_{t+H}^1$ in effect. For instance, in pseudo-labeling approaches, pseudo-labels are regarded as generalized supervisory signals, and numerous studies employ consistency regularization to reinforce the dominance of direct supervision in model learning[2,3], or use thresholding of pseudo-labels to directly discard low-confidence generalized signals [4].  This implies that the signals from direct samples are more reliable than those from indirect samples (pseudo labels). \
> (4) We have included the related discussion in Appendix I in our revised paper.
>
>
> [1] Zhang, Ziqi, et al. "Context-former: Stitching via latent conditioned sequence modeling." arXiv preprint arXiv:2401.16452 (2024). \
> [2] Abuduweili, Abulikemu, et al. "Adaptive consistency regularization for semi-supervised transfer learning." Proceedings of the IEEE/CVF conference on computer vision and pattern recognition. 2021. \
> [3] Wu, Dong-Dong, Deng-Bao Wang, and Min-Ling Zhang. "Revisiting consistency regularization for deep partial label learning." International conference on machine learning. PMLR, 2022. \
> [4] Guo, Lan-Zhe, and Yu-Feng Li. "Class-imbalanced semi-supervised learning with adaptive thresholding." International conference on machine learning. PMLR, 2022.
>
>
> **Q2.** I believe the high-level intuition of this paper's suggestion as a solution makes sense and is generalizable over HRL. I wonder if this can be extended to other base models, not just on HIQL. e.g., HAC, HGCBC, etc., if the impl or idea is too specific to HIQL. \
> **A2.** Thank you for recognizing our work! Of course, the framework of RD-HRL can be applied to other HRL methods.  \
> (1) Following your suggestion, we have transferred the RD-HRL to HGCBC, and the results are as follows. As shown, RD-HGCBC achieves clear improvements on both antmaze-ultra-play and antmaze-ultra-diverse, with a particularly notable gain of 30.4% on antmaze-ultra-play. This demonstrates the applicability of the RD framework to other algorithms.
>
> | Dataset               | HGCBC | RD-HGCBC |
> |-----------------------|-------|----------|
> | antmaze-ultra-diverse | 39.4  | **42.0** |
> | antmaze-ultra-play    | 38.2  | **49.8** |
>
> (2) We have also transferred the RD mechanism to HAC and propose RD-HAC. The results on ant-four-rooms are summarized in Appendix G.12 of our revised paper. As shown, at 1200k steps, RD-HAC achieves a 35.6% improvement, demonstrating the applicability of RD mechanism. (In the first 700k steps, the replay buffer is sparsely populated, making RD mechanism cannot yet fully demonstrate its advantages. After 1500k steps, as exploration progresses, RD-HAC and HAC produce comparable results.)
> Moreover, since HAC is an online RL method, the advantage brought by RD mechanism also suggests that RD-HRL can also offer significant benefits in online RL settings.

---

> > ### Comment · Reviewer_5uMD · 2025-11-27
> >
> > I appreciate the authors' considerable response to my concerns.
> >
> > **Regarding Q1 (unreliable sub-goals)**: My concerns are resolved. The authors explained the concept of unreliable sub-goals technically and referred to concrete supporting evidence with related papers. Furthermore, the authors clearly included these discussions in the Appendix.
> >
> > **Regarding Q2 (generalizability)**: I sincerely appreciate that the authors took my feedback seriously and conducted the suggested experiments. The results demonstrating RD-HGCBC's improvements are compelling evidence that the RD mechanism generalizes beyond HIQL. Also, it's impressive that the RD mechanism can be applied to even online RL setups successfully, although the shown experiments couldn't demonstrate the advantage of the RD approach effectively yet. These significantly strengthen the contribution.
> >
> > **Follow-up question**: I wonder how the authors applied the RD mechanism to the online RL (HAC) framework. Since the main preprocessing phase for extracting a reliable region might only be allowable on offline datasets, I couldn't fully understand RD-HAC's implementation details. It would be very helpful if the authors could share the detailed process of RD-HAC for future work to apply the RD mechanism to online approaches effectively.

---

> > > ### Author Response · Authors · 2025-11-27
> > > **Response to Reviewer 5uMD (Details of RD-HAC)**
> > >
> > > Thank you for your recognition of our work! We are glad that we have addressed your earlier concerns. We are also happy to share the implementation details of RD-HAC with you.
> > >
> > > Firstly, we would like to clarify that the main preprocessing phase for extracting a reliable region can be applied to any collected trajectory, not only to offline reinforcement learning datasets. In online reinforcement learning settings (such as HAC), we can leverage the replay buffer to perform transition regions extraction.
> > >
> > > Specifically, RD-HAC was implemented based on a public Pytorch implementation of HAC, https://github.com/hai-h-nguyen/Hierarchical-Actor-Critic-Pytorch. We retain the three-layer HAC architecture, and introduce the RD mechanism at the top layer. The RD mechanism analyzes the top-layer replay buffer and extracts transition regions.
> > >
> > > The highest layer of HAC maintains a replay buffer of interaction records, which provides the foundational support needed for implementing RD-HAC. However, the replay buffer in HAC stores only (s, a, r) tuples rather than complete trajectory information. Therefore, in RD-HAC, we augment HAC by synchronously supplementing these tuples with their corresponding trajectory information. In this way, we maintain HAC's replay buffer as a list of complete trajectories, which is essentially equivalent to the dataset used in offline reinforcement learning. With this, we can naturally transfer the RD mechanism from the offline reinforcement learning setting into HAC.
> > >
> > > We still need to note that online reinforcement learning continuously interacts with the environment, meaning that the collected replay buffer is constantly changing. Therefore, to ensure that the extracted transition regions to be updated with the explored environment, we adopt an alternating training scheme: after the top layer provides the second layer with T₁ exploration-oriented subgoals using HAC’s original policy, we apply the RD mechanism to generate T₂ reliability-driven decision-level targets (subgoals) for the second layer. In our preliminary implementation, we set T₁ = 500,000 and T₂ = 10,000.
> > >
> > > Overall, our RD-HAC incorporates the following key improvements：
> > > 1. Enhancing the original replay buffer of the top layer in HAC so that it can record full trajectory information.
> > > 2. Taking the trajectory replay buffer as an offline dataset  to apply the RD mechanism.
> > > 3. To ensure that the extracted transition regions to be updated with the explored environment, we adopt an alternating training scheme.
> > >
> > > We hope our responses have addressed the additional questions you raised. If you have any further concerns, please feel free to reach out to us. We look forward to further discussion with you.

---

> ### Author Response · Authors · 2025-11-27
> **Request for Further Discussion**
>
> Dear Reviewer 5uMD,
>
> Thank you for your time and effort in reviewing our paper. We have provided detailed responses to your insightful comments and concerns, and we hope our responses have addressed them. The responses can be summarized as follows:
>
> We have provided a more comprehensive definition of unreliable subgoals, along with additional explanations, which are organized in Appendix I.
>
> In accordance with your suggestions, we further evaluated the effectiveness of the proposed reliability-driven (RD) mechanism on both HGCBC and HAC. The results indicate that our RD  mechanism yields improvements in both behavior cloning and online reinforcement learning settings.
>
> We sincerely hope these clarifications have addressed your concerns. If so, we would greatly appreciate your consideration in updating your rating. If you have any other concerns or questions, please feel free to share them with us. We look forward to further discussions with you.
>
> Best regards, \
> The Authors

---

### Official Review · Reviewer_nnn4 · 2025-11-01

**Soundness:** 2
**Presentation:** 3
**Contribution:** 2
**Rating:** 4
**Confidence:** 3

**Summary:**

- This paper tackles the credit assignment problem in long-horizon, sparse-reward offline reinforcement learning (RL) tasks.
- The authors argue that existing HRL methods are unreliable because the value functions used to select sub-goals suffer from generalization noise.
- The paper proposes Reliability-Driven HRL (RD-HRL).
- This method introduces a reliability-driven decision mechanism composed of three new components
  1. Transition Region Extraction (TRE): K-Means clustering and a "Future Diversity Index (FDI)" to identify "transition regions"
  2. Target Evaluation (TE): A value function trained only on states within these transition regions
  3. Target Identification (TI): A policy that selects a long-term "decision-level target"
- The authors claim this framework provides "noise-immune" guidance, leading to SOTA performance on benchmarks.

**Strengths:**

- Despite some flawed analysis (see "Weaknesses"), the paper presents good results on some tasks. For example, Antmaze-Ultra benchmarks appear to be statistically significant and state-of-the-art
- The paper introduces a new 3-level hierarchy (TI $\rightarrow \pi^h \rightarrow \pi^l$) to tackle the problem of sub-goal generation HRL
- The paper provides an extensive set of ablations (e.g., RD-HRL-TRE, RD-HRL-HP, RD-HRL-TE) that attempt to validate the necessity of each new component

**Weaknesses:**

- The analysis is flawed and contains multiple errors that inflate performance
  - The claim of achieving "the best performance across all [manipulation] benchmarks except for the kitchen-mixed task" is false. Table 1 clearly shows PlanDQ (75.0) and DTAMP (74.4) are superior on kitchen-partial and kitchen-mixed, respectively
  - The claim of a "57%" outperformance over HIQL on CALVIN  is an incorrect calculation and a cherry-picked comparison. The 25-point delta (68.8 vs 43.8) is not 57%, and the more relevant comparison is to the second-place DiffuserLite (52.1)
- The paper's method for claiming statistical wins is non-standard and not well-justified
  - The 3% rule is justified by citing papers that use different rules (e.g., 5% rule in Chen et al and Li et al or CIs in Badrinath et al (see Table 1 row 1 -- halfcheetah))
- The "Future Diversity Index" (FDI) metric is not clearly justified
- There is a typo on line 144: "To improve the the action-level..."
- In Table 3, the CIs for RD-HRL and HIQL on CALVIN clearly overlap

**Questions:**

- The central claim about solving "generalization noise" is confusing. Can the authors provide a precise explanation for how the TE module's "cross-trajectory propagation" is fundamentally different from the "generalized Bellman backup" it claims is the problem? Why is one reliable and the other not, when both operate on the same transition regions?
- The paper claims SOTA on manipulation tasks (except one), but Table 1 shows this is false. Please correct this claim or explain the discrepancy
- Can the authors provide a stronger theoretical or empirical justification for the FDI metric? As written, it seems arbitrary.

---

> ### Author Response · Authors · 2025-11-21
> **Responses to Reviewer nnn4 (Q1-Q3)**
>
> Dear Reviewer nnn4,
> Thank you for your valuable comments. We reorganized the issues into Q1-Q7 for clearer discussion and aligned them with the reviewers’ comments. We use R-Wx to denote Weakness x and R-Qx to denote Question x.
>
> **Q1 [R-W1, R-Q2].** The paper claims SOTA on manipulation tasks (except one), but Table 1 shows this is false. Please correct this claim or explain the discrepancy. \
> **A1.** Thank you for pointing that out! We have revised this sentence as "the top 3% performance across all manipulation benchmarks except for the kitchen-partial task". We have made the adjustments to correct the related typos, please refer to our revised paper for details.
>
> **Q2 [R-W1].** The claim of a "57%" outperformance over HIQL on CALVIN is an incorrect calculation and a cherry-picked comparison. The 25-point delta (68.8 vs 43.8) is not 57%, and the more relevant comparison is to the second-place DiffuserLite (52.1). \
> **A2.** Thank you for your comment. \
> (1) We believe 57% is correct, as (68.8 - 43.8) / 43.8 = 0.570. \
> (2) We believe it is more reasonable to compare RD-HRL with HIQL, as HIQL serves as our base model. Moreover, even compared with the second-place DiffuserLite (52.1), our method still achieves a clear advantage by (68.8 - 52.1) / 52.1 = 32.1%.
>
> **Q3 [R-W2].** The paper's method for claiming statistical wins is non-standard and not well-justified (reasons of using top-3%  rather than top-5%). \
> **A3.** Thank you for your comment. Actually, both top-5% and top-3% are used in previous works; for instance, top-3% has been used in [1]. We adopt 3% to better highlight our advantage, since 3% is more stringent than 5%. To address your concern, we present the results for the top 5% in the following table. It can be observed that by bolding top-5%, our method RD-HRL still achieves the top-5% performance in 8 out of 9 situations.
>
>
>
> | Datasets               | PlanDQ            | MSCP             | V-ADT          | DTAMP            | HD-DA            | HILP          | HILP-Plan     | HIQL           | DiffuserLite    | RD-HRL             |
> |------------------------|-------------------|------------------|----------------|------------------|------------------|---------------|---------------|----------------|------------------|---------------------|
> | antmaze-medium-diverse | **93.0 ± 2.6**    | 88.9 ± 2.2       | 52.6 ± 1.4     | 88.7 ± 3.7       | 88.7 ± 8.1       | 43.5 ± 7.6    | 49.2 ± 5.1    | 86.8 ± 4.6     | 87.6 ± 2.0       | **94.6 ± 2.5**      |
> | antmaze-medium-play    | **92.1 ± 1.7**    | **91.3 ± 1.3**   | 62.2 ± 2.5     | **93.3 ± 0.9**   | 85.8 ± 2.4       | 45.6 ± 4.0    | 46.6 ± 10.4   | 84.1 ± 10.8    | 88.8 ± 3.2       | **94.0 ± 1.2**      |
> | antmaze-large-diverse  | 86.0 ± 3.5        | 83.4 ± 3.2       | 36.4 ± 3.6     | 78.0 ± 8.8       | 83.6 ± 5.8       | 46.0 ± 12.7   | 64.5 ± 10.2   | **88.2 ± 5.3** | 75.2 ± 3.5       | **91.3 ± 4.3**      |
> | antmaze-large-play     | 85.3 ± 6.3        | 86.5 ± 1.1       | 16.6 ± 2.9     | 80.0 ± 3.3       | 80.7 ± 6.1       | 49.0 ± 8.8    | 58.8 ± 11.2   | 86.1 ± 7.5     | 69.4 ± 6.5       | **95.3 ± 2.1**      |
> | antmaze-ultra-diverse  | 70.0 ± 4.5        | 55.1 ± 7.3       | –              | 59.2 ± 3.1       | 52.2 ± 6.9       | 21.2 ± 11.2   | 59.2 ± 12.7   | 52.9 ± 17.4    | 69.3 ± 2.5       | **81.1 ± 6.3**      |
> | antmaze-ultra-play     | **71.5 ± 3.3**    | 36.0 ± 14.3      | –              | 49.9 ± 7.1       | 59.1 ± 5.5       | 22.2 ± 11.4   | 50.8 ± 9.6    | 39.2 ± 14.8    | 63.7 ± 4.2       | **72.9 ± 5.1**      |
> | kitchen-partial        | **75.0 ± 7.1**    | 36.9 ± 3.3       | 46.0 ± 1.6     | 63.4 ± 8.8       | **73.3 ± 1.4**   | 63.9 ± 5.7    | 59.7 ± 5.1    | 65.0 ± 9.2     | **71.4 ± 1.2**   | 69.6 ± 7.4          |
> | kitchen-mixed          | 71.7 ± 2.7        | 44.5 ± 5.3       | 46.8 ± 6.3     | **74.4 ± 1.4**   | 71.7 ± 2.7       | 55.5 ± 9.5    | 51.9 ± 8.3    | 67.7 ± 6.8     | 64.8 ± 1.8       | **72.9 ± 1.7**      |
> | CALVIN                 | 45.0 ± 19.8       | 49.9 ± 11.5      | –              | 51.3 ± 2.9       | 44.6 ± 11.7      | 12.1 ± 5.1    | 14.5 ± 2.5    | 43.8 ± 39.5    | 52.1 ± 1.1       | **68.8 ± 9.7**      |
>
>
> [1] Li, Haoran, et al. "Stabilizing diffusion model for robotic control with dynamic programming and transition feasibility." IEEE Transactions on Artificial Intelligence 5.9 (2024): 4585-4594.

---

> ### Author Response · Authors · 2025-11-21
> **Responses to Reviewer nnn4 (Q4-Q6)**
>
> **Q4 [R-W3, R-Q3].** The "Future Diversity Index" (FDI) metric is not clearly justified. Can the authors provide a stronger theoretical or empirical justification for the FDI metric? \
> **A4.** Thank you for your question. \
> (1) As discussed in Section 1, we refer to regions where multiple trajectories are close to each other as transition regions.
> Theoretically, by connecting more trajectories, transition regions typically admit a larger set of possible future directions; thus, therefore, clusters with more possible future directions are very likely to be transition regions. Hence, we first discretize the dataset through clustering, then identify transition regions by counting the number of future clusters for each cluster and selecting those with greater future possibilities based on the FDI index.  \
> (2) Empirically, to further illustrate the role of FDI in identifying transition regions, we visualize the transition regions selected under different FDI values; please refer to Appendix G.11 in our revised paper for details. In summary, using FDI > 0 as the selection criterion yields more reasonable transition regions. \
> (3) We have also evaluated how different FDI thresholding choices affect model performance to highlight the impact of FDI empirically, and we summarize the results in the table below. As can be observed, as the FDI threshold becomes more permissive, the model’s performance gradually degrades. The drop is particularly pronounced when the (FDI × N + 2) decreases from 3 to 2. This is because most non–transition regions have two future clusters; thus, when we relax the FDI threshold from 3 to 2, the model is exposed to a large number of noisy transition regions, which leads to the observed performance degradation.
>
>
> | FDI * N + 2 | antmaze-ultra-play | antmaze-ultra-diverse |
> |-------------|---------------------|-------------------------|
> | 1           | 56.6                | 62                      |
> | 2           | 60.3                | 64.3                    |
> | 3           | 72.9                | 81.1                    |
>
>
>
> **Q5 [R-W4].** There is a typo on line 144: "To improve the the action-level..." \
> **A5.** Thank you for pointing that out! We have made the adjustments to correct the typos, please refer to our revised paper for details.
>
> **Q6 [R-W5].** In Table 3, the CIs (Confidence intervals) for RD-HRL and HIQL on CALVIN clearly overlap. \
> **A6.** Thank you for your comment. We agree that the CIs for RD-HRL and HIQL on CALVIN clearly overlap. However,  that does not affect our conclusion that RD-HRL performs much better than HIQL. In terms of the mean, RD-HRL achieves a 57% improvement over HIQL; in terms of standard deviation, RD-HRL exhibits a much smaller variance than HIQL, which demonstrates that RD-HRL is more stable than HIQL.

---

> ### Author Response · Authors · 2025-11-21
> **Responses to Reviewer nnn4 (Q7)**
>
> **Q7 [R-Q1].** Can the authors provide a precise explanation for how the TE module's "cross-trajectory propagation" is fundamentally different from the "generalized Bellman backup"? Why is one reliable and the other not, when both operate on the same transition regions? \
> **A7.** Thank you for your question. \
> (1) First, we would like to further explain the concept of the generalized Bellman backup and the reasons for its unreliability. Taking Figure 1 as an example, to estimate the value of $s_{t+H}^1$, the agent needs to generalize the value estimation from $s_{k^2}$ to $s_{k^1}$ to perform the complete Bellman backup from $g$ to $s_{t+H}^1$. This process inherently involves a requirement for generalization, as $s_{k^2}$ and $s_{t+H}^1$ are located in different trajectories, thus there is no path for the direct Bellman backup from $g$ to $s_{t+H}^1$. We refer to such a Bellman backup as a generalized Bellman backup. However, as reported in previous studies [1,2,3,4], in both reinforcement learning and other domains, the generalized signal is often attenuated and unreliable, introducing noise into the value estimation of $s_{t+H}^1$, as illustrated in Figure 1 (c). \
> (2) In our work, we adopt an approximation technique within the transition region to circumvent such generalized signals. Note that in Eq. (8), as we have emphasized in our paper, the state pairs used for the Bellman backup, $(s_{t_1} ,s_{t_2})$, may come from different trajectories. This enables a direct Bellman backup from $s_{t_1} $ to $s_{t_2}$, especially when they are not located in the same trajectory.
> Taking Figure 1 as an example again, with Eq. (8), we can now perform the direct Bellman backup via { ${g}$ → $s_{k_2}$ → $s_{t+H}^1$}, rather than relying on the generalized Bellman backup, thereby avoiding the unreliable value estimation.
>
> [1] Zhang, Ziqi, et al. "Context-former: Stitching via latent conditioned sequence modeling." arXiv preprint arXiv:2401.16452 (2024). \
> [2] Abuduweili, Abulikemu, et al. "Adaptive consistency regularization for semi-supervised transfer learning." Proceedings of the IEEE/CVF conference on computer vision and pattern recognition. 2021. \
> [3] Wu, Dong-Dong, Deng-Bao Wang, and Min-Ling Zhang. "Revisiting consistency regularization for deep partial label learning." International conference on machine learning. PMLR, 2022. \
> [4] Guo, Lan-Zhe, and Yu-Feng Li. "Class-imbalanced semi-supervised learning with adaptive thresholding." International conference on machine learning. PMLR, 2022.

---

> ### Author Response · Authors · 2025-11-27
> **Request for Further Discussion**
>
> Dear Reviewer nnn4,
>
> Thank you for your time and effort in reviewing our paper. We have provided detailed responses to your insightful comments and concerns, and we hope our responses have addressed them. Some of the key clarifications include the following:
>
> Clarification on representations. We have revised the description of the performance on the Kitchen task and corrected other typos in the paper.
>
> Additional ablation studies. We provide both quantitative and qualitative justifications for the FDI metrics in Appendix G.8 and Appendix G.11, respectively. The results demonstrate that the FDI metric we proposed is both reasonable and feasible.
>
> We have cross-checked the calculations related to performance improvements in the paper and can confirm their correctness.
>
> Also, we would like to update the kitchen-mixed row in the table in A3 as follows, because the bold formatting of  results on PlanDQ and HD-DA  in the initial responses did not render correctly:
>
> | Datasets               | PlanDQ            | MSCP             | V-ADT          | DTAMP            | HD-DA            | HILP          | HILP-Plan     | HIQL           | DiffuserLite    | RD-HRL             |
> |------------------------|-------------------|------------------|----------------|------------------|------------------|---------------|---------------|----------------|------------------|---------------------|
> | kitchen-mixed          | **71.7 ± 2.7**        | 44.5 ± 5.3       | 46.8 ± 6.3     | **74.4 ± 1.4**   | **71.7 ± 2.7**       | 55.5 ± 9.5    | 51.9 ± 8.3    | 67.7 ± 6.8     | 64.8 ± 1.8       | **72.9 ± 1.7**      |
>
>
> We sincerely hope these clarifications have addressed your concerns. If so, we would greatly appreciate your consideration in updating your rating. If you have any other concerns or questions, please feel free to share them with us. We look forward to further discussions with you.
>
> Best regards, \
> The Authors

---

### Official Review · Reviewer_DegB · 2025-11-02

**Soundness:** 3
**Presentation:** 2
**Contribution:** 2
**Rating:** 4
**Confidence:** 3

**Summary:**

The paper proposes RD-HRL, a hierarchical offline reinforcement learning method that aims to make sub-goal selection more reliable in long-horizon sparse-reward problems. It first clusters states to extract “transition regions” with a Future Diversity Index, then trains a Target Evaluation (TE) module that estimates values only over these regions and a Target Identification (TI) module that selects a decision-level target (a transition region) given the current state and goal. At evaluation time, TI provides a decision-level target to the high-level policy, which outputs an action-level sub-goal for the low-level policy to execute. Experiments on antmaze, kitchen, and CALVIN benchmarks report strong average performance and ablations analyze the contribution of each component.

**Strengths:**

- Clear problem motivation: The paper articulates how value-function generalization noise can mislead sub-goal selection in hierarchical methods and uses a didactic example to illustrate the issue.
- Architectural novelty: Introducing a layer that constrains high-level decisions to transition regions/bottleneck states and learning a temporally abstracted value function over these regions is an interesting design choice. The split between decision-level targets (TI output) and action-level targets (from the high-level policy) is conceptually clean.
- Broad set of experiments: The method is evaluated on standard and challenging long-horizon benchmarks (antmaze including ultra variants, kitchen, CALVIN), and reports strong results relative to recent HRL baselines such as HIQL (Park et al., 2024a), PlanDQ (Chen et al., 2024a), DTAMP (Hong et al., 2023), and DiffuserLite (Dong et al., 2024), with helpful ablations (RD-HRL-TE, -TRE, -HP, -CU) and a horizon-size comparison.
- Useful ablations and analysis: The ablations disentangle the effects of transition regions, temporal abstraction, and cross-trajectory updates; the horizon-size study clarifies that performance gains are not merely from increasing $H$ in a standard HRL pipeline.

**Weaknesses:**

Notation and parameter-role inconsistencies:
  - Section 2.2 describes $\beta$ as the discount factor even though $\gamma$ is the discount factor throughout the paper. In AWR-style objectives $\beta$ is typically a temperature; this should be corrected to avoid confusion with $\gamma$.
  - The TI objective (Eq. 10) uses $\exp(\beta^{d(s\_t, s\_z)} \cdot A\_{TI})$ together with text stating “raised to the power of $d(s\_t, s\_z)$.” This deviates from the standard AWR form $\exp(\beta \cdot A)$ and from the high-/low-level objectives in Eq. 3 and 4. The paper should justify or correct this exponentiation vs. linear scaling.
- Unclear writing. From a first read, it was very unclear whether $g\_{TI}$ is a region identifier, a representative state from a region, or a learned embedding for the region. The paper would definitely benefit from a few additional iterations of editing/re-writing.
- Cross-trajectory updates in Eq. (8) require a notion of temporal distance $d(s\_{t1}, s\_{t2})$ across trajectories. The approximation used can introduce bias or high variance when trajectories differ in speed/length. Some empirical sensitivity analysis or an uncertainty-aware alternative would strengthen the claim.
- The reliance on K-Means clustering and a fixed FDI threshold may limit generality in high-dimensional or non-Euclidean state spaces. The procgen-500 pilot supports this concern: performance is poor when clustering raw pixels and improves only with external embeddings. This suggests that end-to-end learned representations or graph-based region discovery could be necessary for broader applicability.

**Questions:**

1. Could the authors clarify the intended role of $\beta$ in Eq. (10)? Is it a temperature parameter (as in AWR) or related to the discount factor $\gamma$? If the exponentiation form $\beta^d(s_t, s_z)$ is intentional, what is the theoretical or empirical justification for raising $\beta$ to the power of distance?
2. What exactly does $g_{TI}$ represent at evaluation time - a region index, a representative state, or a learned embedding of the region? How is $\pi_h$ conditioned on this variable in practice?
3. There are mismatches between text, pseudocode, and tables (e.g., Algorithm 1 labels, Table 1 vs. Table 3). Could the authors reconcile these discrepancies and clarify whether they affect experimental results?
4. Since the method relies on K-Means and Euclidean distances, how well does it scale to high-dimensional or non-Euclidean spaces? Have the authors tried graph-based or density-based alternatives, and if so, how do results compare?
5. In Eq. (8), temporal distance $d(s_{t1}, s_{t2})$ is approximated across trajectories. How sensitive are TE’s estimates to this approximation, and does it introduce bias or variance in value learning?
6. The paper includes several ablations, but could the authors include one specifically isolating the impact of the FDI thresholding and clustering scheme to show its independent contribution to performance?
7. How frequently does TI select each transition region during evaluation, and does this selection correlate with TE’s estimated reliability or downstream task success?
8. Why do the authors claim the robotic tasks in CALVIN benchmark are non-euclidean? Could they clarify what makes these non-euclidean?

---

> ### Author Response · Authors · 2025-11-21
> **Responses to Reviewer DegB (Q1-Q4)**
>
> Dear Reviewer DegB, \
> Thank you for your valuable comments. We reorganized the issues into Q1–Q10 for clearer discussion and aligned them with the reviewers’ comments. We use R-Wx to denote Weakness x and R-Qx to denote Question x.
>
> **Q1 [R-W1, R-Q1]**. Could the authors clarify the intended role of $\beta$ in Eq. (10)? Is it a temperature parameter (as in AWR) or related to the discount factor $\gamma$? In AWR-style objectives $\beta$ is typically a temperature; this should be corrected to avoid confusion with $\gamma$. \
> **A1**. Thank you for your question. Following HIQL[1], we subsume the discount factors into the temperature hyperparameter $\beta$ for simplicity. Following your suggestion, we have adjusted the presentation in our revised paper.
>
> [1] Park, Seohong, et al. "Hiql: Offline goal-conditioned rl with latent states as actions." Advances in Neural Information Processing Systems 36 (2023): 34866-34891.
>
> **Q2 [R-W2, R-Q1].** What is the theoretical or empirical justification for raising $\beta$ to the power of distance? The paper should justify or correct this exponentiation vs. linear scaling. \
> **A2.** Thank you for your question. Firstly, as the distance increases, the advantage of a subgoal relative to the final goal diminishes; thus, we must introduce a decay mechanism. Secondly, the idea of exponential decay is inspired by the AWR-style loss in HIQL. In HIQL’s AWR-style loss, the scaling of advantages between adjacent steps is linear (with the discount factor absorbed into β). However, in our method, each pair of adjacent transition regions abstracts multiple regular RL steps, making the overall effect equivalent to the product of linear scalings, which naturally leads to an exponential decay.
>
> **Q3 [R-W3, R-Q2].** What exactly does $g_{TI}$ represent at evaluation time - a region index, a representative state, or a learned embedding of the region? How is $\pi_h$ conditioned on this variable in practice? \
> **A3.** Thank you for your question. \
> (1) The $g_{TI}$ in our paper represents a state in both the training and evaluation phases.  \
> (2) It might be confusing that the symbol $g_{TI}$ is not shown in the learning objective of $\pi_h$, i.e., Eq.(3). Actually, we use $g$ to denote the condition of $\pi_{h}$ in Eq.(3) solely for compatibility with prior HRL methods. In practice, $\pi_h$ of RD-HRL is designed to achieve decision-level targets, thus $\pi_h$ is actually learned on  ${s}_z$, which are states from transition regions. \
> During the evaluation phase, as is described in Eq.(7), the TI module "reconstructs" a proper ${s}_z$,which is known as the decision-level target and is noted as $g\_{TI}$. After that, $\pi_h$ predicts the action-level target conditioned on the current state $s_t$ and the decision-level target $g\_{TI}$. \
> We have added a relevant explanation in our paper to avoid any potential misunderstandings. Please refer to our revised paper for details.
>
> **Q4 [R-Q8].** Why do the authors claim the robotic tasks in CALVIN benchmark are non-euclidean? Could they clarify what makes these non-euclidean? \
> **A4.** Thank you for your comment. In Figure 2 of the official CALVIN paper [1], the authors mention that the proprioceptive state (i.e., the observation space) includes the End-Effector Orientation. Prior works show that the End-Effector Orientation belongs to the rotational component of SE(3), i.e., SO(3)[2,3,4]. Since the rotational component SO(3) is a 3-dimensional smooth manifold with nonzero curvature, it is inherently non-Euclidean [5,6]， making the observation space of CLAVIN non-Euclidean. Consequently, we treat CALVIN, whose observation space contains non-Euclidean features, as a non-Euclidean task.
>
>
> [1] Mees, Oier, et al. "Calvin: A benchmark for language-conditioned policy learning for long-horizon robot manipulation tasks." IEEE Robotics and Automation Letters 7.3 (2022): 7327-7334.\
> [2] Portela, Tifanny, et al. "Whole-body end-effector pose tracking." 2025 IEEE International Conference on Robotics and Automation (ICRA). IEEE, 2025.\
> [3] Berenson, Dmitry, Siddhartha Srinivasa, and James Kuffner. "Task space regions: A framework for pose-constrained manipulation planning." The International Journal of Robotics Research 30.12 (2011): 1435-1460.\
> [4] Murray, Richard M., Zexiang Li, and S. Shankar Sastry. A mathematical introduction to robotic manipulation. CRC press, 2017.\
> [5] Liu, Yulin, et al. "Delving into discrete normalizing flows on so (3) manifold for probabilistic rotation modeling." Proceedings of the IEEE/CVF conference on computer vision and pattern recognition. 2023.\
> [6] Schuck, Martin, Sherif Samy, and Angela P. Schoellig. "A Primer on SO (3) Action Representations in Deep Reinforcement Learning." arXiv preprint arXiv:2510.11103 (2025).

---

> ### Author Response · Authors · 2025-11-21
> **Responses to Reviewer DegB (Q5-Q8)**
>
> **Q5 [R-W5, R-Q4].** The reliance on K-Means clustering and a fixed FDI threshold may limit generality in high-dimensional or non-Euclidean state spaces; an  end-to-end learned representations or graph-based region discovery could be necessary for broader applicability. Since the method relies on K-Means and Euclidean distances, how well does it scale to high-dimensional or non-Euclidean spaces? \
> **A5.** Thank you for your comment. \
> (1) As shown in Table 1, we have demonstrated the superior performance of RD-HRL on the non-Euclidean task CLAVIN; compared to our base method HIQL, RD-HRL achieves a 57% improvement, demonstrating that the K-Means clustering and a fixed FDI threshold adopted by RD-HRL can handle the non-Euclidean state spaces.  \
> (2) We also evaluated the performance of RD-HRL on high-dimensional tasks in Appendix G.1. The results show that the task-relevant embedding leads to performance improvements, also demonstrating the potential of RD-HRL with K-Means clustering and a fixed FDI threshold in high-dimensional spaces. \
> (3) We agree with you that end-to-end learned representations could be necessary for broader applicability. As is discussed in Section 6, we have discussed that developing an end-to-end method based on RD-HRL could further extend its applicability. However, our paper focuses on investigating transition regions and their ability to mitigate generalization noise in long-horizon sparse reward tasks (including high-dimensional and non-Euclidean tasks). Exploring the broader applicability of this work lies beyond its scope and will be addressed in future research.
>
> **Q6 [R-W5, R-Q4].** Have the authors tried graph-based or density-based alternatives, and if so, how do results compare? \
> **A6.** Thank you for your question. We had experimented with density-based clustering DBSCAN [1] and graph-based clustering HDBSCAN[2], but the clustering results turned out to be chaotic, making it impossible for us to infer transition regions or conduct further experiments based on those cluster methods. We believe this is because DBSCAN and HDBSCAN assume that clusters consist of density-connected regions, but trajectory data typically form sparse and non-uniform structures, making density-based assumptions unreliable. \
> We have updated the corresponding clustering visualizations in our revised paper; please refer to Appendix G.6 for details.
>
> [1] Deng, Dingsheng. "DBSCAN clustering algorithm based on density." 2020 7th international forum on electrical engineering and automation (IFEEA). IEEE, 2020. \
> [2] McInnes, Leland, John Healy, and Steve Astels. "hdbscan: Hierarchical density based clustering." J. Open Source Softw. 2.11 (2017): 205.
>
>
> **Q7 [R-Q3].** Mismatches between text, pseudocode, and tables (e.g., Algorithm 1 labels, Table 1 vs. Table 3).  \
> **A7.** Thank you for pointing that out! We have made the adjustments to correct the typos. Please refer to our revised paper for details. These typos do not affect the actual implementation and the experimantal results of RD-HRL.
>
> **Q8 [R-W4, R-Q5].**  In Eq.(8), temporal distance $d(s_{t_1}, s_{t_2})$ is approximated across trajectories. How sensitive are TE’s estimates to this approximation, and does it introduce bias or variance in value learning? \
> **A8.** Thank you for your insightful question! As we select $s_{t_1}$ and $s_{t_2}$ from different trajectories to perform direct Bellman backups, there is no 'golden standard' for $d(s_{t1}, s_{t2})$, and thus there are no proper definitions for bias or variance. \
> However, as is discussed in our paper, we carefully choose a state $s_{t'_2}$ that is similar to ${s}\_{t_2}$  but comes from the same trajectory as $s\_{t_1}$, and take $t'\_{2}$ - $t\_{1}$ as $d(s\_{t_1}, s\_{t_2})$ to approximate the cross-trajectory goal distance as accurately as possible. We have tried other approximate schemes, for example, performing Monte Carlo sampling within the target transition region for approximation. However, the training process with these alternative schemes failed to converge. Therefore, from the empirical perspective, the similarity-based approximation we adopt is a more reasonable strategy. Nevertheless, thank you for the suggestion. We will explore incorporating uncertainty-aware mechanisms in future work to improve the stability of RD-HRL.

---

> ### Author Response · Authors · 2025-11-21
> **Responses to Reviewer DegB (Q9-Q10)**
>
> **Q9 [R-Q6].** The paper includes several ablations, but could the authors include one specifically isolating the impact of the FDI thresholding and clustering scheme to show its independent contribution to performance? \
> **A9.** Thank you for your suggestion!  \
> (1) Following your suggestion, we evaluated how different FDI thresholding choices affect model performance, and we summarize the results in the table below. As can be observed, as the FDI threshold becomes more permissive, the model’s performance gradually degrades. The drop is particularly pronounced when the (FDI × N + 2) decreases from 3 to 2. This is because most non-transition regions have two future clusters; thus, when we relax the FDI threshold from 3 to 2, the model is exposed to a large number of noisy transition regions, which leads to the observed performance degradation. \
> (2) Regarding the impact of the clustering scheme, we explored how different numbers of clusters affect model performance in Appendix G.4. In addition, following your suggestion, we also experimented with density-based clustering methods and graph-based clustering methods; please refer to A6 for details.
>
> | FDI * N + 2 | antmaze-ultra-play | antmaze-ultra-diverse |
> |-------------|---------------------|-------------------------|
> | 1           | 56.6                | 62                      |
> | 2           | 60.3                | 64.3                    |
> | 3           | 72.9                | 81.1                    |
>
> **Q10 [R-Q7].** How frequently does TI select each transition region during evaluation, and does this selection correlate with TE’s estimated reliability or downstream task success? \
> **A10.** Thank you for your question. We computed the probability distribution over the transition regions selected by TI across 3,500 trajectories, and we also measured the normalized score associated with each selected transition region. The results are summarized in Appendix G.10. \
> The results are summarized in Appendix G.10, in which the horizontal axis represents the IDs of the 18 extracted transition regions, labeled sequentially from 0 to 17; the vertical axis shows the probability of being selected after passing through a given transition region and the average Normalized Score obtained after passing through that transition region, respectively. \
> It can be observed that the transition regions chosen by the TI module correspond to high success rates, demonstrating that the TI module’s selection of transition regions is positively correlated with downstream task success. \
> In addition, transition region 14 provides a representative example of a low-return transition region. Its extremely low selection probability indicates that our method is capable of avoiding transition regions that yield low success rates for the current state. In other words, TI’s selection of transition regions is closely related to the success of downstream tasks.

---

> ### Author Response · Authors · 2025-11-27
> **Request for Further Discussion**
>
> Dear Reviewer DegB,
>
> Thank you for your time and effort in reviewing our paper. We have provided detailed responses to your insightful comments and concerns, and we hope our responses have addressed them. Some of the key clarifications include the following:
>
> Clarification on representations. We have adjusted the representations according to your suggestions and corrected the typos in the paper.
>
> Additional ablation studies. Based on your suggestions and questions, we have added ablation experiments on FDI thresholding (Appendix G.8) and the clustering scheme (Appendix G.6) to verify the reasonableness of the FDI values and the use of K-Means clustering.
>
> Additional visualization analysis. We also supplemented visualizations of the transition region selection probability and downstream task performance (Appendix G.10), demonstrating the correlation between the TI module, decision-level targets, and downstream task success rates.
>
> We sincerely hope these clarifications have addressed your concerns. If so, we would greatly appreciate your consideration in updating your rating.
> If you have any other concerns or questions, please feel free to share with us. We are looking forward for further discussions with you.
>
> Best regards, \
> The Authors

---

### Author Response · Authors · 2025-11-21
**General Response**

We would like to express our sincere appreciation to all reviewers for your constructive feedback. We have revised our paper according to your feedback (highlighted in blue in the paper). Specifically, we have made the following changes:
1. We supplement our earlier experiments with clustering results and analyses using the density-based method DBSCAN and the graph-structured method HDBSCAN, further validating the effectiveness of our choice of K-Means. ( see Appendix G.6 )
2. Applying our framework to HGCBC, we demonstrate that the proposed reliability-driven mechanism generalizes well to other methods, further highlighting the significance and originality of our contribution (see Appendix G.7).
3. Through quantitative analysis of the results ( see Appendix G.8 ) and qualitative examination of the filtered transitions ( see Appendix G.11 ), we further validate the impact of the FDI threshold. These findings support the soundness of the FDI metric and the partitioning criterion introduced in Eq.(6).
4. We further provide  visualizations of the transition regions identified by RD-HRL on the manipulation tasks, demonstrating that our method effectively detects Transition Regions in these settings. ( see Appendix G.9)
5. We additionally provide the selection probabilities of decision-level targets and the task success rates associated with the selected targets. The results and relevant analysis show that the selection of transition regions tends to favor those with higher downstream success rates as decision-level targets (see Appendix G.10).
6. We have also transferred RD-HRL to an online RL method, HAC, and proposed RD-HAC. The results show that the RD-HRL framework can also provide significant benefits in online reinforcement learning (see Appendix G.12).
7. We have enriched the discussion of unreliable sub-goals (see Appendix I).
8. We modify the descriptions in the paper to avoid typos and ambiguity.

---

### Author Response · Authors · 2025-12-03
**Final summarisation of author-reviewer discussion**

Dear Area Chairs, Program Chairs and Senior Area Chairs,

We would like to express our sincere gratitude for your tremendous efforts in ensuring a fair and professional review process, especially under the current challenging circumstances. For your convenience, we would like to provide a brief summary of our rebuttal.

# Main Contributions of our work
Our paper proposes RD-HRL, which introduces a reliability-driven (RD) mechanism into conventional HRL. By extracting transition regions and providing proper transition regions as goals for the high-level policy, this mechanism addresses the issue that HRL models often fail into suboptimal subgoals, a problem that is widely observed in HRL settings and often results in degraded performance.


# Overview of Discussion

Our paper was reviewed by four reviewers. We are pleased that we have **addressed most of the concerns of Reviewers 5uMD and XFsY** several days before the leak of reviewers' identities. We are also pleased that **Reviewers DegB and nnn4** consider our paper to have strong advantages, highlighted by its clear **motivation**, **novelty**, **strong performance**, and **thorough ablation studies**.
Although we did not receive responses from Reviewers DegB and nnn4, we provided point-by-point replies to their comments, and we believe our responses sufficiently address their concerns.



# Strengths
We are pleased that the reviewers have acknowledged the following strengths of our work:
1. Clear **motivation** and architectural **novelty**: " Introducing a layer that constrains high-level decisions to transition regions/bottleneck states and learning a temporally abstracted value function over these regions **is an interesting design choice**";  "The authors also introduce a **novel** metric".`DegB`  `nnn4` `5uMD`
2. Strong average **performance**: "Experiments on antmaze, kitchen, and CALVIN benchmarks report strong average performance". `DegB`  `nnn4` `5uMD` `XFsY`
3. **In-depth Analysis**: "The paper provides a thorough ablation study examining each component of the framework".`DegB`  `nnn4` `5uMD`
4. Approach **generalizability**: "This paper offers a **general solution**";  "it's **impressive** that the RD mechanism can be applied to **even online RL setups successfully**".  `5uMD`
5. **Solid overall quality**: "Overall, I find this to be a **solid paper** that clearly demonstrates improvements in offline settings." `XFsY`

# Core Raised Concerns and Our Responses
The main concerns and corresponding responses are summarised as follows:
- Reviewers 5uMD and XFsY wonder about the generalizability of the RD mechanism to other methods (HGCBC, HAC) or settings (behaviour-cloning, online RL). `5uMD` `XFsY`
    - Behaviour-cloning (BC) setting: We combined RD with HGCBC. The results demonstrate that the proposed RD mechanism generalises well to BC settings, highlighting the generalizability of the RD mechanism.  (**Appendix G.7, A2 for Reviewer 5uMD**)
    - Online RL setting: We combined RD with HAC, and the results show that the RD mechanism can also provide significant benefits in online RL settings. (**Appendix G.12, A2 for Reviewer 5uMD, A1 for Reviewer XFsY**)
- Understanding of Eq.(8). `DegB` `XFsY`
    - Reviewer DegB wonders about the reliability of the approximation in Eq.(8).  We have provided the detailed  discussion in A8 for Reviewer DegB. From the empirical perspective, the similarity-based approximation we adopt is a reliable strategy. (**A8 for Reviewer DegB**)
    - Reviewer XFsY wonders about a deeper understanding of why Eq.(8) can overcome generalisation error. Referring to Figure 1, we provided a more detailed explanation in A6 of our response to Reviewer XFsY. Reviewer XFsY stated that our responses satisfactorily addressed his/her concerns. (**A6 for Reviewer XFsY**)
- Reviewers DegB and nnn4 are curious about the impact of the FDI thresholding and clustering scheme (alternative cluster methods beyond KMeans).
    - Justification of FDI metrics. We validate the reasonableness and effect of the FDI metrics via quantitative analysis (Appendix G.8) and qualitative examination (Appendix G.11).  (**Appendix G.8, G.11,  A9 for Review DegB, A4 for Reviewer nnn4**)
    - Alternative cluster methods beyond K-Means. We supplement clustering results and analyses using the density-based method DBSCAN and the graph-structured method HDBSCAN, further validating the effectiveness of our choice of KMeans. (**Appendix G.6,  A6 A9 for Review DegB** )



We sincerely appreciate the time and effort the AC has dedicated to maintaining a fair and balanced review process. Since we have conducted a broad set of additional analyses in this rebuttal, we kindly ask the AC to review our revised paper. The modifications of our revised paper are summarized in **General Response**.

---

### Meta-Review · Area_Chair_yggQ · 2026-01-04

**Summary:**

I recommend accepting the paper. The reviewers appreciated the novelty, strong performance, and in-depth analysis of the work. The scores were slightly negative in the worst case but the author responses to those reviews are thorough and effective.

**Reviewer Concerns:**

Main concerns

Reviewer DegB:
- Has several notation and parameter-role inconsistencies
-  - They seem modest concerns and are addressed by the authors

Reviewer nnn4:
- Claimed that the analysis is flawed
- - The author clarifies that the calculations are correct
- Papers claim of statistical wins not well justified
- - While authors in response provided a more standard way of reporting, a better approach is to run a standard significance test or confidence interval of the mean, since the relevant experiments here already have 50 seeds.

**Reviewer Scores:**

Based on the responses, I would have increased the slightly negative scores to positive ones.

---

### Decision · Program_Chairs · 2026-01-26

Accept (Poster)